

# Multi-annual temperature evolution and implications for cave ice development in a sag-type ice cave in the Austrian Alps

Maria Wind[1,2], Friedrich Obleitner[1], Tanguy Racine[2], and Christoph Spötl[2]

[1]Department of Atmospheric and Cryospheric Sciences, University of Innsbruck, Innsbruck, 6020, Austria
[2]Institute of Geology, University of Innsbruck, Innsbruck, 6020, Austria

**Correspondence:** Maria Wind (maria.wind@uibk.ac.at)

**Abstract.** Ice caves are, similar to mountain glaciers, threatened by the warming climate. To better understand the response of perennial ice in caves to a changing climate, we analysed the thermal characteristics of a sag-type ice cave in the Austrian Alps (*Hundsalm* ice cave), based on long-term temperature measurements for the period 2008–2021. Observations show a warming trend in all parts of the cave as well as a distinct seasonal pattern with two main regimes, i.e., an open (winter) and a closed
(summer) period. During the closed period, a persistent stable stratification prevails that largely decouples the cave from the external atmosphere. The open period is characterised by unstable to neutral stratification and allows episodic penetrations of cold air from outside into the cave interior. Vertical temperature profiles also provide hints on corresponding circulation patterns and the spatial temperature variability in the cave. The positive air temperature trend is reflected in a decrease in perennial cave ice, derived from stake measurements. Besides surface melting, we find compelling evidence of basal melting
of ice. The observed ablation rates can be well reproduced by applying a modified degree-day model, which, however, is less feasible regarding mass balance. Overall, we conclude that *Hundsalm* ice cave is highly impacted by regional warming which will lead to the disappearance of its perennial ice deposits within the next decades.

## 1 Introduction

Hidden below the Earth's surface, ice caves represent a small but fascinating part of the cryosphere. Per definition, ice caves are
"rock-hosted caves containing perennial ice or snow, or both" (Luetscher and Jeannin, 2004b). Similar to mountain glaciers, their larger surface counterparts, ice caves are threatened by global warming. Unlike glaciers, however, underground ice deposits can be located far outside the boundaries of the permafrost zone in areas with mean annual air temperatures well above 0 °C (Perşoiu et al., 2011; Obleitner and Spötl, 2011), rendering them particularly vulnerable to any warming trend (Kern and Perşoiu, 2013). The recently observed decrease in cave ice deposits (e.g., Kern and Thomas, 2014; Perşoiu et al., 2021; Securo
et al., 2022) highlights the urgency to reinforce efforts of studying these subterranean archives.

Differentiation between ice caves is usually based on the origin of cave ice (snow-derived firn or in situ formed congelation ice) and cave air dynamics (Bögli, 1980; Luetscher and Jeannin, 2004b) both of which are determined by the cave morphology (e.g., number and elevation of cave entrances, vertical and horizontal extent). Sag-type caves represent one idealised end member of the cave morphology spectrum. Having one or more entrances at similar elevation and showing a (near-)vertical



geometry, they are characterised by a natural, seasonally controlled ventilation pattern. In the summer months (closed period) the cave atmosphere is largely decoupled from the outside while in the winter months (open period) the cave serves as a thermal trap for cold air (Perşoiu, 2018).

The scientific interest in sag-type ice caves is linked to the great potential of their perennial ice deposits for the reconstruction of past climate and environmental change, reaching back several hundreds to locally a few thousand years (e.g., Gradziński 30 et al., 2016; Luetscher et al., 2007; Munroe, 2021; Spötl et al., 2014). For a robust interpretation of these archives, however, it is crucial to assess and understand the microclimatic and glaciological conditions inside ice caves and their coupling to the outside atmosphere. In this respect, studying temperature already yields major insights as it incorporates information about relevant processes controlling the cave climate. The thermal disequilibrium between the cave and the outside was found to control the rate and direction of airflow (de Freitas et al., 1982; Faimon and Lang, 2013; Meyer et al., 2016), which is a major 35 player in the energy balance of a cave system (Luetscher et al., 2008). The strong correlation of positive temperature sums with several components of the energy balance furthermore serves as the basis for modelling glacier melt rates and has been widely applied in glaciological studies (e.g., Braithwaite, 1995; Hock, 2003; Kuhn et al., 1999; Lang and Braun, 1990). Although temperature is most commonly measured in ice caves around the world, studies are mostly based on short time periods leaving a gap for long-term investigations of ice cave temperature. Furthermore, the spatial distribution and temporal consistency of 40 these measurements are mostly insufficient to allow comprehensive analyses.

We aim to fill this gap by using long-term temperature measurements and stake records at a sag-type ice cave in the Austrian Alps, *Hundsalm* ice cave, ranging from 2008 to 2021. Similar to other well studied ice caves e.g., *Eisriesenwelt* (Obleitner and Spötl, 2011; Schöner et al., 2011), *Dachstein Rieseneishöhle* (Saar, 1956) and *Scărişoara* ice cave (Racovita and Onac, 2000), *Hundsalm* ice cave is not a purely natural system because it is used as a touristic show cave. Most human interference is, 45 however, well documented and the main analyses of this work are not affected. Otherwise, studying special cases such as the winter 2011/12 when the door to the cave accidentally fell shut, can help to better understand the cave dynamics and investigate their response to disturbances (natural or artificial). Corresponding results are not only of scientific interest but also important for regional tourism with regard to show cave management and preservation of this special habitat. Previous studies of this cave already showed snapshots of the data presented here, but focused on other aspects like seasonal ice growth (Spötl, 2018) 50 and past climate reconstruction (Spötl et al., 2014). In this work, we performed a thorough analysis of the thermal conditions at *Hundsalm* ice cave over more than one decade, elaborating on average conditions as well as spatial and temporal temperature variations. We further investigated the link of the cave atmosphere to the outside environment by means of vertical profiles and stability analysis. Finally, we explored the relationship between temperature (in- and outside the cave) and cave ice dynamics and address the potential of modelling cave ice evolution based on temperature information. Results of this study should serve 55 as a scientific baseline for further, more detailed studies of the cave's micrometeorology as well as future modelling approaches.





## 2 Data and methods

### 2.1 Field site

*Hundsalm* ice cave (hereafter HIC) is located in the western part of the Northern Calcareous Alps (47° 32' 42" N, 12° 01'
35" E) at an elevation of 1520 m above sea level. This sag-type cave opens to the outside via two quasi-vertical shafts (referred
to as upper and lower entrance) and consists of an upper (ice-bearing) level and a lower (ice-free) part, separated by an artificial
airlock. The upper entrance is a 25 m high shaft, while the lower entrance, which opens a few meters below the upper one,
serves as the main entrance and is equipped with a staircase as well as a door (15 m below the upper entrance) that is closed
during summer and fall. The two shafts are connected via a sub-horizontal passage about 12 m below the surface. The main
chamber (*Eisdom*) of the upper level is approximately 42 m long and up to 10 m wide and its bottom is located 34 m below the
upper entrance (Fig. 1).

The lower, ice-free part of the cave extends to 55 m below the entrance and is characterised by a relatively constant tem-
perature of approximately 4 °C. After the discovery of the lower cave level an airlock was installed to prevent air exchange
between the two cave parts (Spötl et al., 2014).

Perennial firn and ice deposits are present in most parts of the upper level except the southernmost part, near *Tropfsteinhalle*.
The major ice deposit is located in *Eisdom* and reaches a maximal thickness of 4 to 5 m, which decreases towards the south. In
the northern part of the cave, below the lower entrance, mainly snow and firn deposits are present. From the *Eisdom* southward,
the ice body consists of a mixture of granular ice derived from firn and congelation ice that formed by freezing of seepage
water (Spötl et al., 2014).

### 2.2 Measurements

Continuous monitoring at HIC started in 2005, when three temperature loggers were installed at *Tiefster Punkt* (T36), *Tropf-
steinhalle* (T30) and *Eisdom* (T29), referred to as the main monitoring sites (Fig. 1, lower panel). Since 2007 additional loggers
were installed during various measurement campaigns along the shaft leading to the upper entrance (T6 to T24) and below and
above the entrance door (T18 and T11, respectively) with the numbers referring to the distance below the upper entrance.

The temperature of the limestone bedrock was measured in a horizontal borehole next to the temperature measurements in
*Eisdom* at a depth of 50 and 126 cm (T50r and T126r) with continuous monitoring lasting from November 2012 to March
2017. Ice temperature was measured from September 2016 to February 2020 close to *Tiefster Punkt*, drilled 45 cm deep into
the ice, reaching the interface to the bedrock (T45i).

An automatic weather station (AWS) measured temperature, pressure and humidity outside the cave, 5 m next to the upper
cave entrance, from 2008 to 2015. In 2016 outside air temperature measurements have been continued on a tree next to the
AWS. Overlapping measurements in the period from 29 September 2014 to 02 November 2015 show a good correlation and
yielded a small temperature correction of $\Delta T = -0.088\,°C$. When considering the long time series, the outside stations'
records are merged and referred to as $T_{out}$.



**Figure 1.** Plan view of HIC with stake locations (upper panel) and elevation view with temperature logger locations (lower panel). The blue coloured loggers captured only part of the measuring period, at various intervals between the years 2009 and 2021.



All temperature measurements were performed using HOBO Water Temp Pro data logger with a quoted accuracy of $0.2\,°\text{C}$ from $0\,°\text{C}$ to $50\,°\text{C}$ and a resolution of $0.02\,°\text{C}$. Each logger was calibrated before installation with reference to $0\,°\text{C}$ in an ice water bath. With the exception of ice temperature, all data were recorded at different intervals between two hours and thirty minutes but were homogenised to a $2\,\text{h}$ interval for further analysis. The individual temperature time series were occasionally interrupted due to the failure or replacement of single loggers (significant data gaps: $T_{out}$ March to July 2009, June to September 2014 and November 2015 to June 2016; T36 September 2017 to June 2018). The densest logger network without major data gaps is available for the years 2011 to 2014.

Only discontinuous precipitation data are available from the cave site. Therefore, monthly precipitation sums from a total-isator operated by the Austrian Hydrological Service (station *Buchacker*, located less than $2\,\text{km}$ south-west of the cave) were used.

The cave ice development has been monitored using stakes placed in different parts of the ice body that were measured manually at least twice a year starting in summer 2007 (Fig. 1, upper panel). Over the course of the years some of the stakes suffered damage or melted out. The longest continuous time series of ice height is available from stake B in *Eisdom* for the years 2007 to 2017, at which point it was renewed (and continued as stake B*). In addition to the stake readings the distance between the ice surface and the metal bridge of the tourist trail above was measured in *Eisdom* from 2016 to 2021 (P4). The stake records are complemented by occasional measurements of near-surface firn density.

### 2.3 Anthropogenic influences

Since HIC is opened as a show cave during summer, it is partly manipulated by human activities. Almost every winter local cavers shovel snow into the cave in order to conserve the ice deposit and improve the cave's ice mass balance. Although these activities are well documented, proper quantification of the effect of the artificial snow input on the cave ice mass balance is not feasible. Visiting tourist parties are led through the cave every year from May to October. During this period the door at the lower entrance of the cave stays closed while during the winter months it is kept open to allow air exchange, except for early January 2012, when the door accidentally fell shut and was reopened in April. Although the closed door certainly limits the cave ventilation along the lower entrance (cf. 4.3), some air exchange is still possible through gaps and fissures as well as through the upper entrance shaft. Similar is true regarding intermittent plugging of one or the other entrance by snow. As this paper focuses on the natural conditions in HIC, winter 2011/12 was excluded from most of the analyses. However, the impact of this event as well as other artificial measures on the cave micrometeorology are treated in the discussion as they provide opportunities to explore the cave's response to perturbations.

### 2.4 Data processing

During the period from May to October, when the show cave is open for tourists, short-lived peaks in cave air temperature were observed at loggers installed within a few metres of the walking path, caused by groups of visitors passing the sensors. Affected sensors are T29, T30, T36, T18 as well as the two lowest sensors in the upper entrance shaft (T21 and T24). The peaks were filtered by setting all values within a $24\,\text{h}$ window, that are greater than the median value of this window, to the median



value. The impact of this filter on the average thermal conditions is, however, small with a change in the mean temperature between May and October below $0.02\,°C$ at all affected loggers. Further visual investigation of the data was carried out to remove remaining spikes in temperature throughout the whole year, e.g., by touching the loggers during read-out. To allow for consistent analyses, all data were finally homogenised to $2\,h$ measurement intervals at full hours using the closest measurement
value to the defined date.

Three major gaps (March to July 2009, June to August 2014, November to May 2016) in the outside temperature record ($T_{out}$) were filled separately by linear regression (ordinary least squares) using data from *Hahnenkamm* station, operated by the Austrian national weather service (ZAMG), over multiple years for the respective missing months ($R^2$ for the regression models: 0.94, 0.90 and 0.91; RMSE: $1.96\,°C$, $1.59\,°C$, $1.89\,°C$). The data gap at the cave temperature logger T36 in winter
2018/19 was not filled.

For stability analysis along the entrance shafts, the potential temperature $\theta$ [K] was calculated using pressure measurements at the AWS. Pressure at the respective depths of the loggers was calculated according to the barometric formula assuming a constant moist-adiabatic temperature gradient (Bergmann and Schaefer, 2001). Subsequently, $\theta$ can be calculated by

$$\theta = T \left( \frac{p_0}{p} \right)^{\frac{R}{c_p}} \tag{1}$$

with $p_0 = 1000\,hPa$ and the specific heat at constant pressure $c_p = 1004\,J\,kg^{-1}\,K^{-1}$ and the ideal gas constant for dry air $R = 287\,J\,kg^{-1}\,K^{-1}$. Sensitivity studies using the equivalent potential temperature (assuming a relative humidity of $100\,\%$ at T29) yielded no relevant differences regarding the average vertical profiles and corresponding stability analyses.

For relating cave ice developments with temperature, a degree-day approach was applied. The basic model sets ablation (*abl*) in relation to the positive degree-day sum (*PDD*) using a degree-day factor (*DDF*) that can be found empirically (Braithwaite,
140 1984):

$$abl = DDF \cdot PDD. \tag{2}$$

The *PDD* is calculated as the sum of daily mean temperatures $\geq 0\,°C$. For further analysis freezing degree days (*FDD*) were likewise calculated as the sum of daily mean temperatures $< 0\,°C$. The degree-day sums were calculated separately for the accumulation and ablation periods. As the number of stake readings per year is limited, the ablation and accumulation periods
of the individual years were defined by the availability of stake measurements in spring (April to July) and autumn (September to December) of that year and their duration therefore differs from year to year. Nevertheless, the measurements were chosen in a way that represents the natural periods as closely as possible. Finally, degree-day sums were normed by the number of days in each period.

*Python3* was used for processing and analysing the data, with the packages *pandas* for time series analysis (resampling,
averaging and basic statistics) (The pandas development team, 2020), *statsmodels* for statistics and regression analysis (Seabold and Perktold, 2010), *pymankendall* for calculating trends (Hussain and Mahmud, 2019), and *matplotlib* and *seaborn* for plotting (Hunter, 2007; Waskom, 2021).

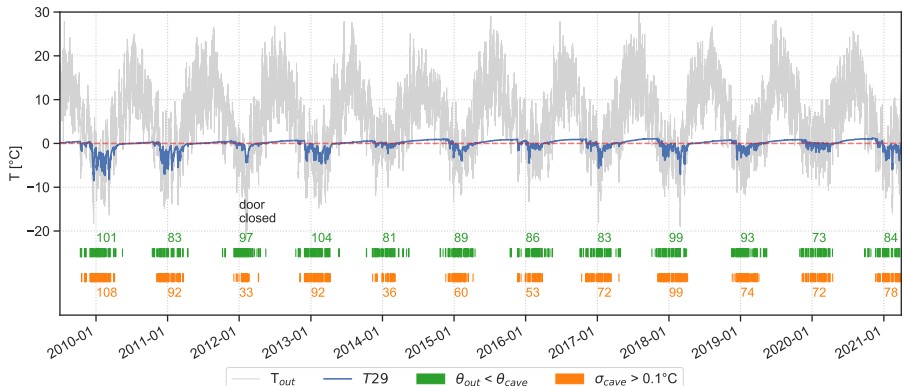

**Figure 2.** Air temperature measured at T29 (*Eisdom*) in blue and next to the cave entrance (grey). Green markers indicate periods when the daily mean potential temperature outside falls below that at T29, orange markers indicate days when the daily standard deviation at T29 exceeds $0.1\,°\mathrm{C}$. The numbers next to the markers indicate the number of days the respective criteria were met for each year. The red dashed line marks the $0\,°\mathrm{C}$ line.

## 3 Results

### 3.1 Climatological conditions

The mean annual air temperature at the cave site is $4.8\,°\mathrm{C}$ (2008–2020) with a mean annual precipitation of $1497\,\mathrm{mm}$ (2009–2018 measured at *Buchacker* station). Key values for the cave temperature are summarised in Table 1. On average, January and February are the coldest months outside and inside the cave. A phase shift can be observed between the interior of the cave and the outside regarding the warmest month. July is the warmest month outside the cave with an average temperature of $13.1 \pm 1.4\,°\mathrm{C}$ and is also the month with the largest temperature difference between the cave and the outside environment.

In contrast, the highest temperatures inside the cave (at the three main monitoring sites) are recorded in November. Applying a Mann–Kendall (Mann, 1945) test to the daily mean temperature shows a significant increasing trend ($p < 0.05$) at the three main cave monitoring locations (T29, T30, T36). The calculated trend is strongest in *Eisdom* (T29) with a rate of $0.054\,°\mathrm{C\,yr^{-1}}$ and decreases with increasing distance to the entrance ($0.049\,°\mathrm{C\,yr^{-1}}$ at T30 and $0.027\,°\mathrm{C\,yr^{-1}}$ at T36). For the outside station, no robust trend could be calculated for the available period. In the lower level of the cave, the temperature is relatively constant

with a value of $4.24 \pm 0.10\,°\mathrm{C}$ for the period of 2008 to 2019. The small deviations from the mean, however, record a seasonal pattern (with a minimum value in February ($4.19 \pm 0.09\,°\mathrm{C}$) and a maximum in April ($4.29 \pm 0.11\,°\mathrm{C}$) as well as an overall positive temperature trend of $0.024\,°\mathrm{C\,yr^{-1}}$.

As expected for sag-type caves (cf. Luetscher and Jeannin, 2004a), two main regimes/seasons can be observed at HIC. During the open (winter) season lasting approximately from December to March, the cave air temperature drops below $0\,°\mathrm{C}$

with frequent and sharp negative temperature excursions closely following the external temperature (Fig. 2). During the rest of the year (closed/summer period), when the temperature outside rises above the temperature inside the cave, temperature



**Table 1.** Average climatological parameters outside HIC as well as at the three main monitoring sites inside the cave (2009–2020). All temperatures are given in $^\circ$C. $\Delta T$ is the mean temperature difference between outside and the respective location inside the cave.

| logger | $T_{annual}*$ | $T_{max}$ | $T_{min}$ | $\Delta T_{Jul}$ | $\Delta T_{Jan}$ |
|--------|---------------|-----------|-----------|------------------|------------------|
| $T_{out}$ | $+4.81 \pm 0.78$ | $32.17$ | $-21.97$ | | |
| $T29$ | $-0.16 \pm 0.48$ | $1.32$ | $-8.54$ | $12.57 \pm 1.38$ | $-1.94 \pm 1.45$ |
| $T30$ | $+0.68 \pm 0.35$ | $1.43$ | $-3.27$ | $12.05 \pm 1.37$ | $-3.56 \pm 1.68$ |
| $T36$ | $-0.27 \pm 0.36$ | $0.66$ | $-5.88$ | $12.89 \pm 1.39$ | $-2.60 \pm 1.58$ |

*years 2017 and 2018 for T36 are excluded from the calculation of the annual mean due to multiple
months of missing data.

variations in the cave are muted and the temperature slowly rises towards $0\,^\circ$C. April and November are considered transition months, incorporating characteristics of both open and closed periods, depending on the outside conditions. This seasonal temperature pattern is similar in all three main cave monitoring sites with the two stations closest to the ice body (T29 and
T36) showing consistently lower temperatures than T30.

To further distinguish between the open and closed season inside the cave, different criteria can be defined, incorporating the thermal characteristics of the open period. Rapid temperature changes are an indication of active air exchange between the cave and the external atmosphere and hence for the open period of the cave. Therefore, the days with a standard deviation greater than $0.1\,^\circ$C ($\sigma > 0.1\,^\circ$C) inside the cave are chosen as a first criterion (orange markers in Fig. 2). This threshold was exceeded
for 105 days during winter 2009/10. The winters of 2011/12 (closed door) and 2013/14 stand out with only 33 and 36 days meeting this criterion, respectively. The second, and also the most basic criterion for the open period is for the potential air temperature outside to drop below that inside the cave ($\theta_{out} < \theta_{cave}$). This signals unstable stratification, enabling the intrusion of cold outside air into the cave (green markers in Fig. 2). This criterion was met during 73 days in the winter half-year of 2019/20 and for 104 days in 2012/2013 at logger T29. For cave logger T29, 77 % and 88 % of the days fulfilling the first and
second criteria, respectively, lie between December and March. If the transition months November and April are included, this number rises to 99 % and 92 %, respectively. However, not all days at which the second criterion was met may induce a significant variation in air temperature inside the cave, as detected by the first criterion. Therefore, we use the first criterion to investigate further how large a temperature difference between the external and the cave atmosphere is required to induce such rapid changes in temperature at different locations in the cave. To induce a daily standard deviation greater than $0.1\,^\circ$C as well
as a net cooling (daily mean temperature change $<0\,^\circ$C) a mean daily temperature difference ($\Delta\theta = \theta_{out} - \theta_{cave}$) of -8.5 $^\circ$C for T30, -6.4 $^\circ$C for T36 and -3.5 $^\circ$C for T29 is required. Minimum values for $\Delta\theta$ are -3.7 $^\circ$C, -1.5 $^\circ$C and +3.6 $^\circ$C for T30, T36 and T29, respectively.





## 3.2 Interannual variability

Looking at the different winter (November to April) and summer (May to October) conditions over the 12 year monitoring
period shows a strong interannual variability not only outside but also inside the cave (Fig. 3). In the first three years of the
monitoring period the winters are characterised by a strong temperature variability and low temperatures at all three cave
stations. At *Tropfsteinhalle* (T30) these were the only years during which the mean winter temperature dropped below $0\,°C$.
The thermal conditions in the cave during winter 2011/12 were different from the preceding and following years as the cave
ventilation was restricted by the unintentionally closed door at the lower entrance from January to April. Although the outside
conditions were comparable to the year before, the temperature inside the cave stayed relatively warmer and only short negative
temperature excursions caused by a strong cold spell in February 2012 were recorded (indicated by the outliers in Fig. 3).
Presumably, this anomalously weak winter cooling had implications for the following summer, when a higher temperature was
recorded inside the cave, compared to years before 2012. The natural influence of a warm winter on the cave air temperature in
the following summer can also be observed after the warm winter of 2013/14 (median $T_{out} = 0.55\,°C$) during which the cave
hardly cooled down. At T30 and T36 especially, the thermal conditions remained similar to the previous summer conditions
and the median temperature stayed above $0\,°C$ at all three cave loggers, marking winter 2013/14 the warmest inside the cave
of the entire observation period. Subsequently, in summer 2014 the cave temperature rose to a median temperature of $0.77\,°C$
at T29, higher than during all previous years. In the following years, except for winter 2017/18, warm winters outside the cave
were followed by even warmer summers inside the cave, with the highest temperatures reached in summer 2017 and 2020 (Fig.
3).

## 3.3 Vertical air temperature profile and stability analysis

To investigate the exchange between outside air and cave air, vertical temperature profiles for the period of June 2011 to May
2014, when additional loggers were installed along the shafts of the upper and lower entrances, were analysed. This increased
logger density provides more insights into the spatial variation of air temperature at HIC. Average monthly temperature varia-
tions for the respective period are shown in Fig. 4. From December to April the median temperature lies below $0\,°C$ at almost
all loggers inside the cave, except for T40 and T30 (Fig. 4). The temporal temperature variability inside the cave is higher
compared to the rest of the year as well, with more variability above $15\,m$ depth (T15). In February the temperature inside and
outside the cave is at its lowest and shows the largest variability. The relatively low variability of T6 in March and April as well
as its temperature close to $0\,°C$ can be explained by the fact that this logger is covered by snow during this time. From April to
November the mean outside temperature exceeds the cave temperature. A strong temperature decrease with increasing depth,
most pronounced in up to $9\,m$ depth in May and up to $18\,m$ in September, leaves the cave atmosphere decoupled from the
outside atmosphere. Hence, in lower parts of the shaft of the upper entrance as well as inside the cave (T18 to T40) this period
is characterised by temperature variations of very small amplitude (no larger than $0.2\,°C$ per day). Early and late cold spells in
April, October and November cause negative temperature deviations (outliers in Fig. 4). Notably, the temperature recorded at
loggers located in the shaft of the lower entrance (T11 and T18) is lower compared to similar elevations in the upper entrance



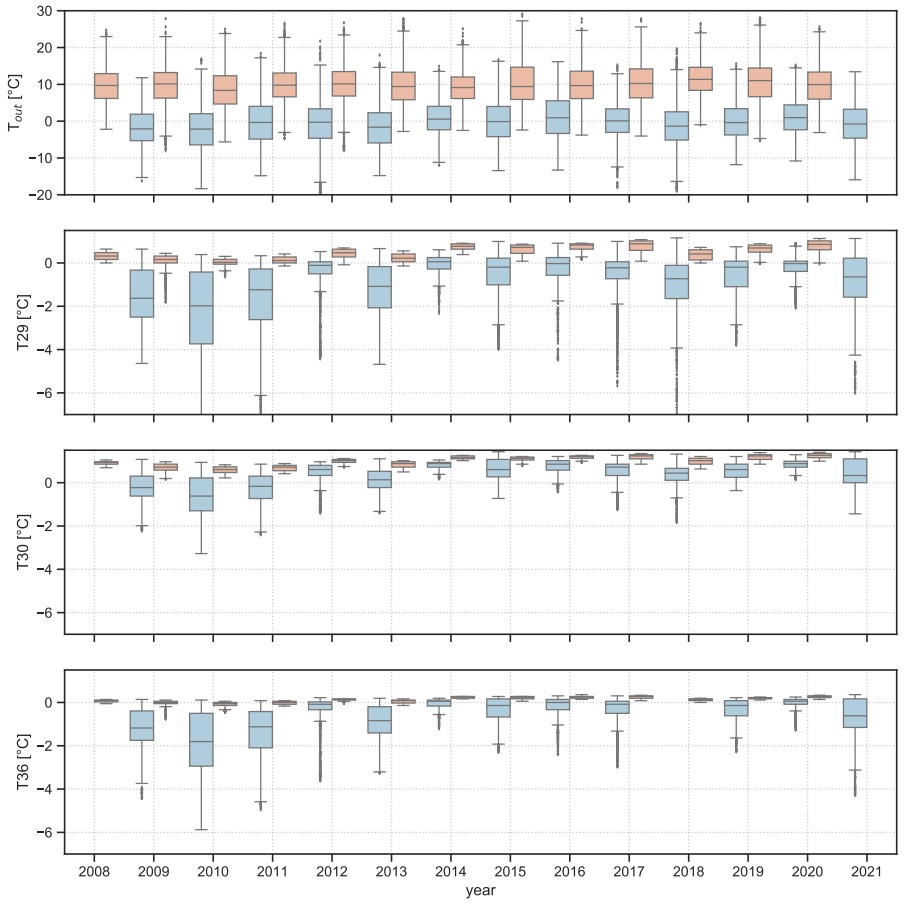

**Figure 3.** Seasonal temperature for the measuring period of June 2008 to March 2021 outside the cave (top) and the three long-term loggers inside the cave for the summer (May to October) and winter (November to April) half-year in orange and blue, respectively. The boxes mark the interquartile range (IQR) and whiskers the distance of 1.5 times the IQR. No box is plotted for a season with more than 2 months of missing data at the respective logger.

shaft (T13, T15, T21) and shows a larger variability. A similar observation was made at two of the main stations inside the cave (T29 and T36) where the temperature is lower than in the lowest part of the upper entrance shaft (T21 and T24). On the other hand, T30 records warmer temperatures throughout the year compared to T29 and T36 and has a lower temporal variability.

Using the temperature characteristics at the logger locations found in the analysis above, the data were separated into two groups: lower and upper entrance. The former group includes T11 and T18 as well as T29 and T36 as they all show stronger temporal temperature variability during the open period. In turn, the upper entrance group includes the loggers placed along the upper entrance shaft (T6, T9, T13, T15, T21, T24) as well as T30 and T40. The vertical profile of potential temperature for the period of June 2011 to May 2014 (Fig. 5) shows the thermal characteristics along the two shafts for the summer and winter period in the cave. During the summer a stable stratification prevails in the shafts below both cave entrances down to a depth



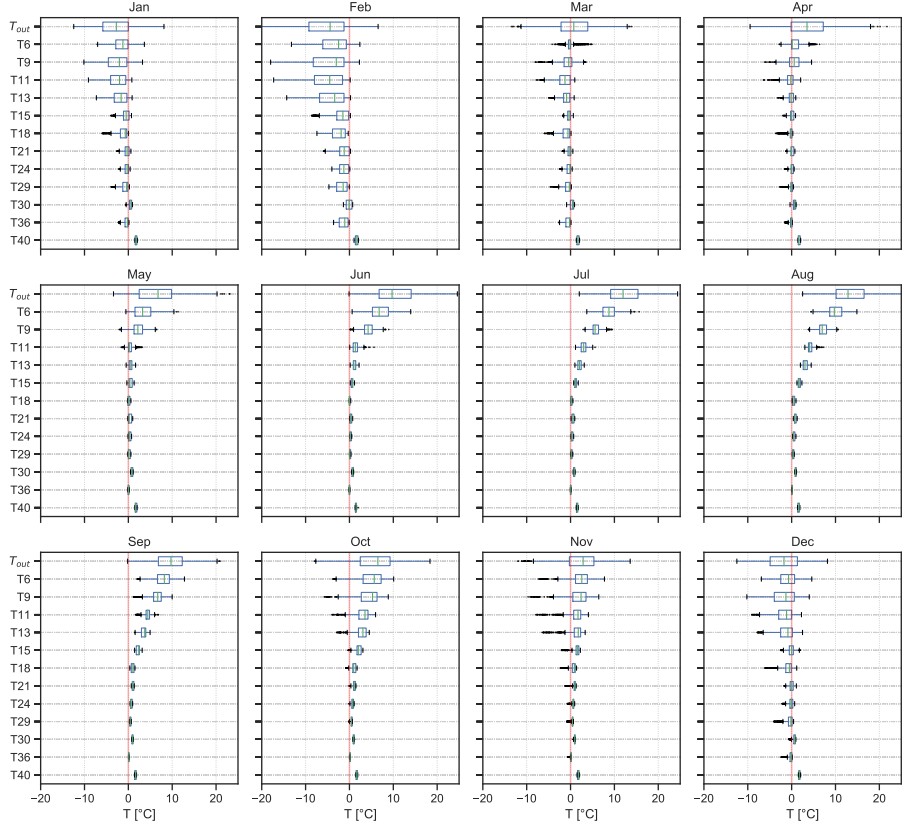

**Figure 4.** Monthly temperature variations for the period of July 2009 to May 2014. The different temperature loggers are plotted along the y-axis from the top down, starting with the outside station ($T_{out}$). The vertical red line marks the $0\,°C$ line. The boxes mark the interquartile range (IQR), whiskers mark the distance of 1.5 times the IQR, the green line denotes the median.

of about $20\,m$. Below, in the main chamber of the cave, slightly stable conditions remain in the ice-bearing (northern) part of the cave (T29 i.e., *Eisdom*). The southern part, leading towards *Tropfsteinhalle* (T30 in Fig. 1), however, shows an unstable stratification during summer as well as during winter. The average winter conditions show a well-mixed to weakly unstable stratification in the shafts below the lower and upper entrance. The temperature along the lower entrance is lower compared to the upper entrance, especially below $12\,m$ depth with a difference of $1\,°C$ at $20\,m$ depth.

The analysed period includes three very different winters regarding the outside temperature, and the intensity and frequency of cold air intrusions into the cave. These different external conditions led to significant differences in the vertical temperature profile inside the cave. The winters 2012/13 and 2013/14 show the strongest contrast with mean outside air temperatures of -4.10 °C and +0.28 °C, respectively. Consequently, the temperature difference between upper and lower entrance was much more pronounced in winter 2012/13 (up to $2\,°C$ difference at a depth between $15\,m$ and $20\,m$). Furthermore, during this winter,

unstable stratification below the lower entrance prevailed from the top down to $36\,m$ depth, enabling air exchange between the



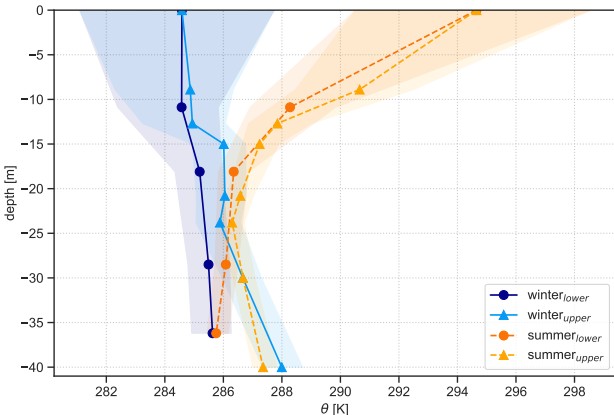

**Figure 5.** Averaged potential temperature profiles over the period of June 2011 to May 2014 for the summer (dashed) and winter (solid) along the upper and lower entrance. Shaded areas around these lines denote the 25 % and 75 % quantiles at the respective logger height. Markers indicate the loggers at the respective depth inside the cave along the lower (triangles) and upper (dots) entrance.

cave and the outside atmosphere. In the following winter of 2013/14, relatively warm conditions outside the cave led to a slightly stable layer in the first 12 m of both shafts. Below that depth, neutral to weakly unstable conditions prevailed. The lack of cold outside air in this winter even led to a warming of the main chamber of the cave, compared to the preceding summer. Comparing the temperature profiles for the summer periods shows that very stable stratification in the upper section of the entrance shafts occurred in each of the analysed years.

### 3.4 Single event characteristics

To scrutinise the response of the cave atmosphere to cold spells outside, the cold air event in January 2013 was studied in more detail (left panel of Fig. 6). This period is characterised by a sharp initial drop in temperature on 10 January of 10 °C and increasingly negative temperatures over the following nine days. During this cooling phase the cave rapidly switched from a closed regime with stable temperature stratification to an open regime where the temperature gradient was reversed. Some loggers, however, did not record the expected pattern of such cold air events with progressively higher temperatures deeper into the cave. Logger T6 located highest in the upper entrance shaft stayed warmer than the loggers below (T9, T11) and recorded largely the same temperature as T15. The temperature at *Tropfsteinhalle* (T30) was the highest during this event and showed only a slight decrease towards the peak of the event. The temperature at the sensor below the door (T18) as well as T29 (*Eisdom*) was lower than in the lower section of the upper entrance shaft (T15, T21, T24) for the duration of the cold spell. A similar cold air event with comparable outside forcing was observed in January 2012. The situation inside the cave, however, was different as the door along the lower entrance fell shut (right panel of Fig. 6). The most striking difference between the two cases is the weaker cooling below the door (T18) of only -1 °C from the beginning to the peak of the event compared to -5.7 °C in the case of 2013. Similarly, the temperature at *Eisdom* (T29) and *Tiefster Punkt* (T36) decreased less during

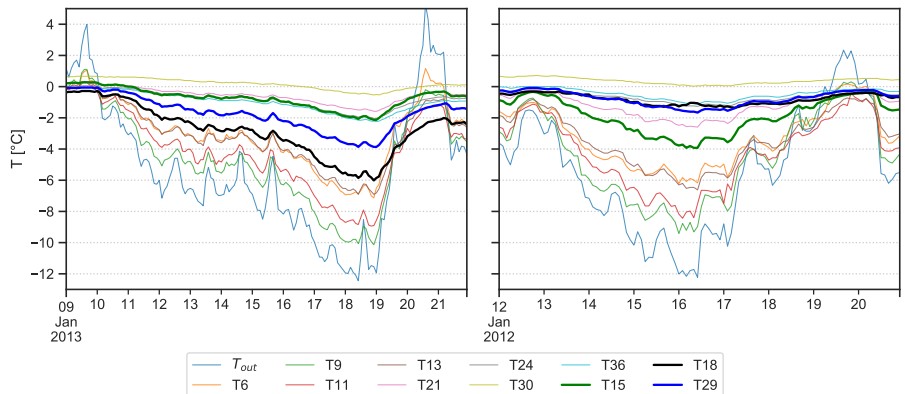

**Figure 6.** Temperature conditions during cold air events in January 2013 (left), showing unrestricted ventilation conditions, and in January 2012 with restricted ventilation due to the closed door at the lower entrance shaft. Bold lines highlight the loggers at which a drastic change in behavior between the two scenarios was observed (T15, T18 and T29).

this event (-1.6 °C vs. -3.8 °C and -1.1 °C vs. -2.2 °C, respectively). On the other hand, the temperature decrease was more pronounced in the lower section of the upper entrance shaft (T15 and T21) compared to the open door situation. In both cases T6 stands out as it stayed warmer than T9 and T13, although it is located closest to the outside station.

### 3.5 Rock and ice temperature

Simultaneous measurements of rock and ice temperature are available for the period 16 September 2016 to 30 March 2017. Similar to the seasonal temperature pattern of cave air temperature, both the rock temperature at 50 and 126 cm depth (T50r and T126r) and the ice temperature (T45i) record negative excursions during the open period and fairly constant temperatures in the closed period (Fig. 7). Compared to the air temperature at T29 the average rock temperature increases and the temporal variability (standard deviation) decreases with depth (T29 = -0.32 ± 1.20 °C, T50r = 0.14 ± 0.69 °C, T126r = 0.27 ± 0.57 °C). The ice temperature shows the lowest average and most variable values overall (-0.32 ± 0.54 °C). Rock temperature measurements show a response to short cold periods in November and December (recorded by T29). Meanwhile, the first notable drop in ice temperature is recorded in January, shortly after the beginning of a long and more intense cooling of the cave. A phase shift between cave air and rock/ice temperature is visible. Cross-correlation with T29 (with a resolution of 6 h) yields a maximum correlation of 0.83 and 0.63 achieved with a lag of 48 and 246 h of T50r and T126r, respectively. For ice temperature, a maximum correlation of 0.87 is reached with a lag of 72 h. The relationship between ice temperature and T36 is even stronger with a maximum correlation of 0.96 at a lag of 48 h. Concerning basal melting, the average length of the period during which the ice temperature stayed at 0 °C was analysed for three full years (2017 to 2019). During this period the temperature stayed at 0 °C for 202 days per year on average.

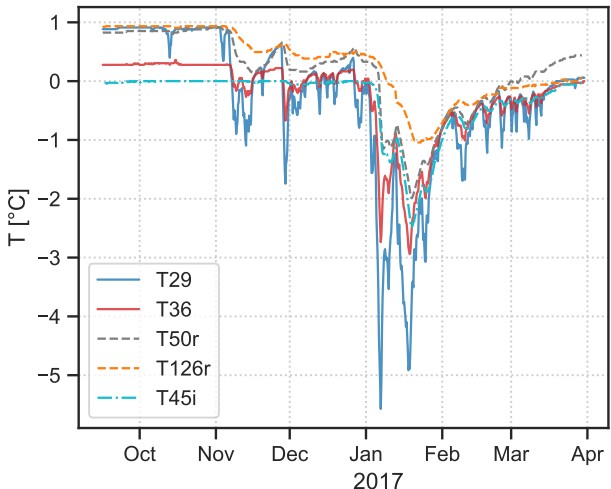

**Figure 7.** Time series of cave air temperature (T29, T36), rock temperature at 50 cm and 126 cm (T50r and T126r) measured in *Eisdom* as well as ice temperature at 45 cm depth at the rock–ice interface (T45i) measured close to *Tiefster Punkt* for the period 2016/09/16 to 2017/03/30 (the only period with parallel measurements of all loggers mentioned above).

### 3.6 Cave ice developments

An overall negative trend in the volume and height of perennial cave ice was observed in all parts of the cave over the last 10
years, with an apparent acceleration since 2014 (Fig. 8). The first distinct ablation period within the observed time frame was in 2012 when almost 20 cm of ice was lost within 166 days at stakes A and B located in *Eisdom* (see Fig. 1 for their locations). Even stronger ice loss was observed after the ablation period of 2014, with a retreat of 30 cm at stake B within 179 days. For accumulation, however, there are only two periods when the ice accumulation exceeded the subsequent ablation at least in parts of the cave (2011 at stake D and 2013 at stake B).

For loggers inside the cave, a strong correlation between the positive degree-day sum (*PDD*) during the ablation periods and the ice height change at stake B (longest continuous measurement series) was found, yielding values for the correlation coefficient *r* of 0.92 for T36 and T30 and 0.93 for T29. The correlation with $T_{out}$ yielded a *r*-value of 0.55. This provides the opportunity to study the potential of a degree-day model, commonly used in glacier mass balance studies (e.g., Hock, 2003; Kuhn et al., 1999), that quantitatively relates ablation to the temperature sum, inside a cave. We use the longest continuous
observation series at stake B (2009–2017) as a reference for the cave ice development and temperature at T29 as the closest and hence most representative measurement for the thermal conditions at the upper boundary of the ice body (cf. Fig. 1). The degree-day factor (*DDF*) was calculated using a linear regression of the *PDD* for the ablation period at T29 and the corresponding melt rate stake B, yielding a value of $1.88 \pm 0.29\,\mathrm{mm\,°C^{-1}\,day^{-1}}$ ($R^2 = 0.84$). Based on the derived *DDF*, average ablation rates (height changes) for the summer periods 2009–2017 were calculated as $12.2\,\mathrm{cm\,yr^{-1}}$ compared to the
observed value of $12.4\,\mathrm{cm\,yr^{-1}}$.





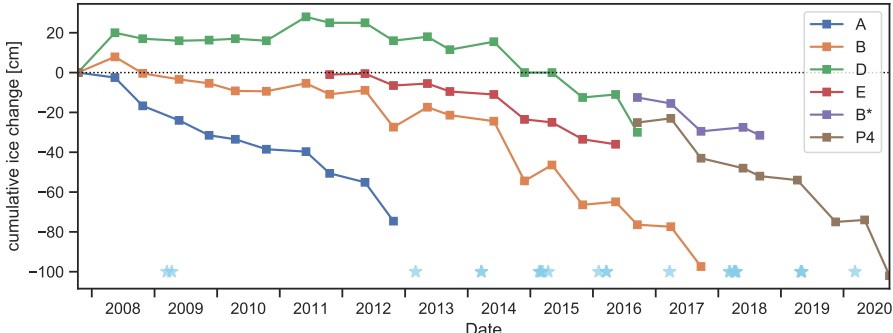

**Figure 8.** Cumulative ice height change at different stakes and along the path (P4) in HIC starting from 2007/10/12. Light blue stars below mark days when shovelling of snow into the cave was documented.

**Table 2.** Properties of the linear regression models for ablation ($abl$) and annual mass balance ($mb$) calculated from degree-day sums at loggers T29 (model T29), T30 (model T30) and $T_{out}$ (model $T_{out}$). For each of the regression coefficients (a, b, c) as well for $R^2$ the values resulting from the model and the 95 % bootstrapping confidence intervals (in brackets) are shown. The RMSE is calculated for the out-of-sample prediction for the years 2008 and 2018–2020, verified with measurements at stakes B (2008) and P4 (2018–2020).

| Ablation | Formula | a | b | | $R^2$ | RMSE [mm day$^{-1}$] |
|---|---|---|---|---|---|---|
| model T29 | $abl = a + b \cdot PDD_{abl}$ | -0.05 [-0.21 0.21] | 1.88 [1.30 2.37] | | 0.86 [0.61 0.98] | 0.33 |
| model T30 | $abl = a + b \cdot PDD_{acc}$ | -0.22 [-0.48 0.13] | 1.92 [1.21 2.52] | | 0.84 [0.54 0.98] | 0.40 |
| model $T_{out}$ | $abl = a + b \cdot FDD_{acc}$ | 1.91 [1.28 2.31] | 0.50 [0.25 0.78] | | 0.65 [0.11 0.91] | 0.50 |

| Mass balance | Formula | a | b | c | $R^2$ | RMSE [mm day$^{-1}$] |
|---|---|---|---|---|---|---|
| model T29 | $mb = a + b \cdot PDD_{abl} + c \cdot FDD_{acc}$ | 0.14 [-0.22 0.59] | -1.21 [-3.84 0.01] | -2.32 [-6.59 6.97] | 0.74 [0.54 0.99] | 0.50 |
| model T30 | $mb = a + b \cdot PDD_{acc} + c \cdot FDD_{acc}$ | 0.96 [-0.22 2.69] | -2.79 [-4.86 -1.08] | 1.02 [-0.56 5.31] | 0.8 [0.60 0.99] | 0.26 |
| model $T_{out}$ | $mb = a + b \cdot FDD_{acc}$ | -1.84 [-2.61 -0.93] | -0.52 [-0.96 -0.17] | | 0.57 [0.10 0.95] | 0.40 |

Further investigation showed that on top of the *PDD* during the ablation period, the *PDD* during the previous accumulation period at *Tropfsteinhalle* (T30) is also closely related to the ablation at stake B. A model fitted with the temperatures measured


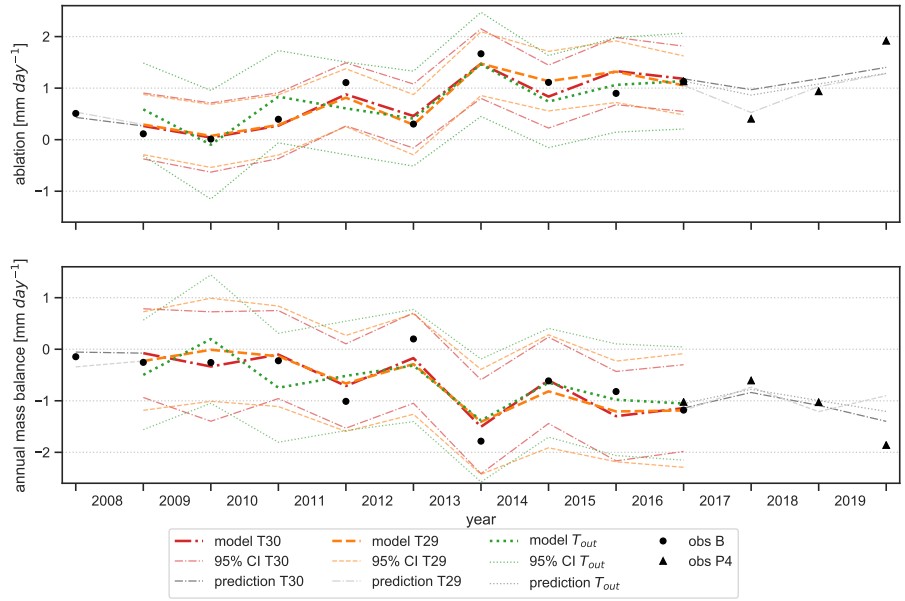

**Figure 9.** Linear regression models of ablation at stake B. Model T29 uses positive degree days during the ablation period at T29 as predictor, model T30 uses positive degree days during the preceding accumulation period as predictor, model $T_{out}$ the freezing degree days during the preceding accumulation period. Lower panel: Linear regression model of the annual mass balance at stake B. Model T29 combines positive degree days during the ablation period and freezing degree days in the preceding accumulation period as predictors, model T30 uses both positive and freezing degree days during the accumulation period as predictors, model $T_{out}$ uses freezing degree days during the accumulation period as a single predictor. Both panels: The years for which the models were fitted (2009 to 2017) are plotted in colour, thin dashed/dotted lines above and below the model line denote the 95 % confidence intervals (CI) of each model. Grey lines show the out-of-sample prediction of the models, the different black markers denote observations at stakes B and P4. For key metrics of the models refer to Table 2.

at the southern edge of the ice body (T30, model T30) yields a similar degree-day factor of $1.92 \pm 0.32 \, \mathrm{mm} \, ^{\circ}\mathrm{C}^{-1} \, \mathrm{day}^{-1}$ ($R^2 = 0.84$) and a similar value of $12.2 \, \mathrm{cm} \, \mathrm{yr}^{-1}$ for the mean ablation. Furthermore, the available data allow us to explore
the use of the degree-day approach to reproduce ablation of cave ice as a function of outside temperature. Using the freezing degree-day sum (*FDD*) of the preceding accumulation period at the outside station yields a *DDF* of $0.5 \, \mathrm{cm} \, \mathrm{yr}^{-1}$. Figure 9 (upper panel) demonstrates that while the performance is similar to models with cave temperature, the confidence interval is, however, considerably larger.

    The lower panel of Fig. 9 shows results of the application of the degree-day model to derive the development of the cave ice
mass balance at HIC. In principle, this necessitates the consideration of mass input (accumulation), which in a cave environment derives from different sources (direct precipitation input, refreezing of seepage water). However, no significant relationship to the outside precipitation (at *Buchacker* station) was found. Other corresponding data are not available for HIC. Nevertheless, examining the performance of multi-linear regression models revealed that the inclusion of the temperature of the preceding





winter as an additional predictor provides some added value. The models for the cave ice mass balance, combining the best

available predictors for accumulation and ablation measured at stake B, are:

– model T30 using *PDD* and *FDD* of the accumulation period at *Tropfsteinhalle* (T30) as predictors,

– model T29 using *PDD* of the ablation period and *FDD* of the accumulation period at *Eisdom* (T29) as predictors and

– model $T_{out}$ using *FDD* of the accumulation period as a single predictor (no significant improvements were achieved using other degree-day sums based on outside temperature) (Fig. 9).

Ablation and mass balance models were fitted for the years 2009 to 2017 and tested using the years 2008 and 2018 to 2020. For the latter, only measurements of P4 are available for verification. Bootstrapping (1000 samples) was applied to determine the 95 % confidence interval for the regression parameters. Key values of each fitted model are shown in Table 2.

### 3.6.1    Basal melting

The ice changes measured by stake readings only account for the changes at the ice surface and therefore give a minimum

value of ice retreat. A second aspect that has to be taken into account is melting at the base of the ice body. Ice temperature measurements close to *Tiefster Punkt* for the years 2017–2019 show a constant temperature at 0 °C for at least June to November, suggesting melting conditions at the base of the ice during approximately six months of the year (Fig. 7). An estimate of basal melt was based upon a period with optimum data coverage (12 May 2016 to 31 March 2017) using three different methods. Firstly, parallel measurements of surface melt at *Eisdom* (stakes B* and B) and total melt (P4, Fig. 1) allow us to

determine the basal melt based on observations. Secondly, the degree-day method was applied to the T36 temperature (using $DDF = 1.88 \, \mathrm{mm} \, °\mathrm{C}^{-1} \, \mathrm{day}^{-1}$ of model T29). Thirdly, the amount of basal melting $m$ through heat conduction at the rock–ice interface was assessed by applying

$$m = \frac{n\lambda \frac{dT}{dz}}{L_{ice}\rho_{ice}} \tag{3}$$

with the latent heat of ice $L_{ice} = 334 \, \mathrm{kJ \, kg^{-1}}$ and n being the number of seconds in the calculation interval. We assumed an

effective thermal conductivity of homogeneous limestone of $\lambda = 2 \, \mathrm{W \, m^{-1} \, K^{-1}}$ and a density of ice of $\rho_{ice} = 920 \, \mathrm{kg \, m^{-3}}$. To calculate the temperature gradient $dT/dz$ we took the mean temperature difference of rock temperature at 50 cm and 126 cm depth. For an alternative estimation, we calculated $dT/dz$ with T36 and T50r with the additional requirement that T36 > 0.2 °C as a measure for potential melt at the air–ice interface. Observations at stakes B* and B suggest basal melting of 7.5 cm and 10 cm in the specified period. The degree-day method as well as Eq. 3 using T36 and T50r yield comparable results of 9.1 cm

and 8.2 cm, respectively. Finally, Eq. (3) applied to the rock temperatures gives the lowest value of 3.2 cm.





## 4   Discussion

### 4.1   Thermal conditions

The presented results paint a comprehensive picture of a sag-type ice cave that is threatened by increasing temperatures. The calculated temperature trends suggest that the average air temperature will rise above $0\,°C$ at *Tiefster Punkt* (T36) and

consequently in the whole ice-bearing level of HIC in 10 years; at *Eisdom* (T29) positive temperatures will be reached in approximately 3 years. For the outside temperature the Mann–Kendall test did not yield a significant trend. This is most probably caused by the relatively short time series used for trend calculations. Using a longer time series (1993–2021) of the highly correlated ZAMG station *Hahnenkamm* results in a trend of $+0.043\,°C\,yr^{-1}$. Assuming that the long-term trend at our cave site is similar means that the warming trend in the main chamber of the cave ($+0.054\,°C\,yr^{-1}$) is even larger compared to

the outside trend.

Looking closer at the interaction between external and cave air we found significant spatial variability inside the cave. Analysing the average temperature difference between the external station and the cave loggers ($\Delta\theta$) necessary to induce temperature variations inside the cave ($\sigma > 0.1\,°C$), shows that *Eisdom* (T29) is most sensitive to changes in the external temperature. At this site, even short cold air excursions lasting less than a day (indicated by an average outside temperature above that of T29) can lead to significant temperature variations. At the deepest point (T36) $\Delta\theta$ needs to be almost $3\,°C$

larger on average compared to T29 to detect similar temperature variations. *Tropfsteinhalle* (T30) is least affected by external temperature variations as it is furthest away from the cave entrances and thus only detects major cold air intrusions when $\Delta\theta \geq \text{-}3.7\,°C$.

The general ventilation characteristics of the cave, specifically the open and closed periods, also found in other sag-type caves

(e.g., Belmonte-Ribas et al., 2014; Luetscher and Jeannin, 2004a; Munroe, 2021), have direct implications for the correlation between the temperature of the external atmosphere and that of the cave air: this correlation must change systematically over the course of a year. From May to September the correlation between $T_{out}$ and *Eisdom* (T29) (monthly Pearson correlation $r$ of $2\,h$ data, 01 June 2008 to 31 March 2021) drops below a value of 0.2 as the outside air is decoupled from the cave atmosphere due to the prevailing stable stratification (Fig. 10, left panel). In contrast, the period from November to March

shows the strongest correlation with values above 0.6 and a maximum correlation of 0.77 in February. *Tiefster Punkt* (T36) and *Tropfsteinhalle* (T30) show a similar seasonal pattern but weaker correlations during the open period (maximum correlation in February of 0.65 at T36 and 0.57 at T30) as a result of the greater distance to the cave entrance compared to T29. Here again, the higher correlation of T36 with $T_{out}$ compared to T30 implies that the connection between the atmosphere outside and in the cave is stronger along the lower cave entrance. The seasonal cycle of correlation shows that the external winter conditions

have a large impact on the cave atmosphere. However, the external temperature variations during winter not only influence the open period but have implications for the subsequent closed period as well.

This relationship between winter and summer temperature inside the cave becomes apparent in the right panel of Fig. 10 where a clear linear relationship can be established between the accumulated temperature during the winter half-year (November to April) and the subsequent summer temperature sum (May to October). The slope of the regression line varies between



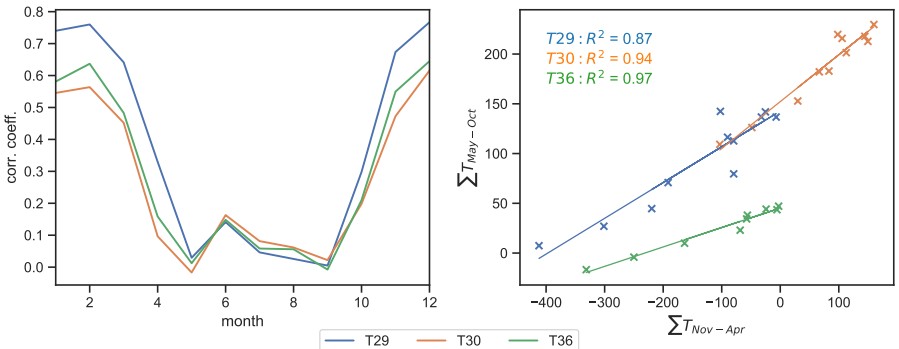

**Figure 10.** Left: monthly mean correlation of cave air temperature (T29, T30, T36) with external temperature ($T_{out}$). Right: cumulative temperature for the accumulation period (November–April) vs. ablation period (May–October); data points are accompanied by a regression line for each of the three main monitoring sites in the cave.

the logger locations with T30 showing the steepest and T36 the lowest incline. The different slopes can be interpreted as the sensitivity of the system at different locations within the cave. A warm winter seems to have the least impact on the deepest point (T36), a place that is surrounded by rock and ice, acting as a thermal buffer. On the other hand, a higher temperature sum in winter is followed by considerably higher temperature sums in the subsequent summer.

## 4.2 Ventilation patterns

Although not directly measured, the main ventilation patterns in HIC can be derived from the temperature observations. The analysis of monthly temperature variations (Fig. 4) for the winter months demonstrates that temperature along the lower entrance (T11, T18) is systematically lower relative to the upper entrance shaft (T15, T21, T24) and exhibits higher temporal variability. This indicates that cold air mainly enters the cave through the lower entrance. On the other hand, the relatively warm temperature especially at T6 but also at T21 and T24 points towards warmer air being pushed out of the cave along the shaft leading to the upper entrance. Regarding the estimation of wind speed, the method proposed by Meyer et al. (2016), to use the phase shift in the temperature data at different logger locations to calculate the air flow velocity, could not be applied in our case. The available $2\,\mathrm{h}$ measuring interval proved to be too coarse as no apparent phase shift can be seen during cold air incursions (Fig. 6). The loggers inside the cave react to the outside temperature drop within the $2\,\mathrm{h}$ interval, hence only a lower limit of the wind speed of $0.004\,\mathrm{m\,s^{-1}}$ can be calculated.

## 4.3 Effects of the open/closed door

Winter 2011/12, during which the door along the lower entrance fell shut, provides the unique opportunity to investigate the response of the cave to a sudden change in the cave geometry fundamentally affecting its ventilation pattern. Although this change was man-made, this scenario can also be thought of in a natural context (e.g., rockfall or snow plugging an entrance). The comparison of the two cold air incursions in January 2012 and 2013 (Fig. 6) reveals different dynamics for the two





scenarios. When the door is open, the temperature at the sensor below the door (T18) as well as in *Eisdom* (T29) is lower than
      at the same elevation of the upper entrance shaft (T15, T21, T24) for the duration of the cold spell, indicating that cold air
      mainly enters through the lower entrance. When this air circulation is restricted due to the closed door, the cold air is diverted
      to the upper entrance shaft, as indicated by the lower temperatures at T15 and T21 and even T29 compared to T18. During
      winter 2011/12, although the cold air still reached the main chamber through the upper entrance, the total cooling inside the

cave (recorded at T29, T30 and T36) was less pronounced. Hence, this unintended experiment demonstrates that the cooling
      mechanism is less effective when the ventilation pattern is restricted to a single entrance. These results highlight the importance
      of proper cave management measures, whereby changes in cave geometry due to e.g. locking or opening of (new) entrances
      can have a profound impact (positive as well as and negative) on the delicate cave environment (c.f. Yang and Shi, 2015). This
      also holds for natural processes e.g., intermittent plugging of cave sections by ice, as observed in ice caves elsewhere (Wimmer,

405   2008).

## 4.4   Cave ice

The observed ice loss at HIC during the monitoring period is consistent with historical evidence showing a gradual long-term
decline in ice volume since the discovery of this cave in 1921 (Spötl, 2013), which accelerated in the past decades. This parallels
trends of negative mass balance in caves around the globe where perennial ice deposits are lost at an increasing rate (Luetscher,

2005; Colucci et al., 2016; Colucci and Guglielmin, 2019; Kern and Perşoiu, 2013; Perşoiu et al., 2021). There were two
significant ablation periods in the first half of the observation period: summer 2012 and summer 2014. The strong melting in
summer 2012 can be attributed to the fact that the cooling of the cave during the previous winter was unusually weak due to the
door at the lower cave entrance being closed (average temperature in the open period 2011/12: $T_{out}$ = -3.2 °C, T29 = -0.7 °C;
closed period 2012: $T_{out}$ = 8.2 °C, T29 = 0.4 °C). Summer 2014, on the other hand, followed a generally weak winter outside

and inside the cave (open period 2013/14: $T_{out}$ = 0.3 °C, T29 = -0.2 °C; closed period 2014: $T_{out}$ = 7.0 °C, T29 = 0.7 °C).

    From the markers in Fig. 8, it is apparent that ice accumulation changes cannot solely be attributed to natural processes. Snow
has been shovelled into the cave on an almost yearly basis in late winter in an attempt to preserve the cave ice. While the dates
when the shovelling took place are documented, no quantitative information of the amount of snow input is available. As far as
possible, any artificially introduced snow was removed from the ice surface when taking the stake measurements. Despite these

protective measures by the show cave management, a negative mass balance was recorded in HIC at every measuring location
since 2013. This anthropogenic influence is also one of several factors that complicates the modelling of the ice developments
in HIC, with the accumulation part proving to be more complex compared to ablation. Efforts to include external precipitation
did not improve the model as no straightforward relationship between outside precipitation and the amount of water entering
the cave can be expected. Due to the rather small diameter of the entrance shafts, the amount of snow or rain falling directly

into the cave is limited. Furthermore, seepage water certainly plays a role in the ice mass balance at HIC too. Due to a lack of
corresponding data e.g., drip water, a quantification of its contribution to the accumulation is not feasible.

    On the other hand, the first-ever application of a degree-day approach to quantify ablation inside an ice cave showed promis-
ing results. Since the concept of relating positive degree days with ablation has never been applied to caves before, the literature





does not provide any reference values. However, for glaciers, a range of values for the *DDF* from 2.5 to $11.6\,\mathrm{mm\,^\circ C^{-1}\,day^{-1}}$

over snow and 5.5 to $20\,\mathrm{mm\,^\circ C^{-1}\,day^{-1}}$ over ice were reported (Hock, 2003, and references therein). The value derived for

HIC using regression analysis ($1.88 \pm 0.29\,\mathrm{mm\,^\circ C^{-1}\,day^{-1}}$) is at the lower end of this range but still in a good agreement

given the fundamental difference of the environments above and below the surface and hence the different relative importance

of processes controlling the energy balance and thus, contributing to ablation. The most obvious is the lack of short wave

radiation in cave environments. In contrast, the long wave radiation, that is already known to be the most important component

of the energy balance for melting on glaciers (Ohmura, 2001), becomes even more so. Regarding turbulent fluxes, it is thought

that they only play a minor role in a cave environment due to turbulence being suppressed by the overall stable stratification

and low velocity of air flow.

Due to the special thermal conditions inside HIC, with summer cave air temperature being strongly influenced by the pre-

ceding winter and a strong correlation between outside and cave temperatures during the open period (Fig. 10), we were able

to extend the classical degree-day approach and relate the ablation also to temperatures during the preceding accumulation

period, hence, making a prediction for summer ablation based on winter conditions. This was not only done with cave air tem-

perature but also with outside temperature, thus establishing a direct connection between cave ice developments and external

temperature albeit with a larger uncertainty compared to cave air temperature. We are aware that the regression algorithms are

used at the limit of their applicability since we are working with very low sample sizes caused by the limited number of years

with continuous observations. To gauge the reliability of the models, bootstrapping was applied, revealing a wide confidence

interval for the regression parameters. The sign and magnitude of the coefficients related to ablation appear to be robust while

the accumulation parts are subject to a higher degree of uncertainty (Table 2). Human interference aside, this increased un-

certainty can be explained by the fact that accumulation is not only temperature-driven but also depends on water availability.

Hence, the comparably good model fit for the annual mass balance can be traced back to the good performance of the model

for ablation which apparently drives mass balance changes. Provided that sufficient observation data are available to fit such a

model, we are confident that this approach can be transferred to other sag-type caves and eventually also can be extended to

model (simplified) long-term cave ice evolution.

The application of the degree-day method to estimate basal melting also showed encouraging results, yielding values com-

parable to the observations in the order of $10\,\mathrm{cm}$ for the year 2016/17. Data from *Tiefster Punkt* (T36) was used for this purpose

not only for the degree-day method but also in the thermal conduction equation because this logger is located closest to the

base of the ice. Furthermore, the base of the ice body likely rests partially on solid rock but also on scree, enabling air to access

the basal ice. This hypothesis is supported by the small amount of basal melting ($3.2\,\mathrm{cm}$) when considering solely the rock

temperature to determine the heat flux provided to the base of the ice. Using the mean gradient between T50r and T36 yields a

result closer to the observations. Limiting basal melting to days when $\mathrm{T36} > 0.2\,^\circ\mathrm{C}$ should furthermore account for the lower

temperatures close in the base of the ice.

Similar values for basal melting were found in Monlési Ice Cave ($8\,\mathrm{cm\,yr^{-1}}$ - Luetscher, 2005), whereas values in Dobšinská

Ice Cave and Scărisoara Ice Cave are significantly lower with only 1 and $1.5\,\mathrm{cm\,yr^{-1}}$, respectively (Tulis and Novotný, 2003;

Perşoiu, 2005). A direct comparison with these literature values is difficult as they are already more than 15 years old and





therefore predate the systematic observations at HIC. Spötl et al. (2014) stated that there was no direct evidence of present-day
basal melting at HIC at the time of their study. For the first years of measurements (2008–2011) the surface melt rate was
smaller with $3.5\,\mathrm{cm\,yr^{-1}}$ at stake B compared to $19.4\,\mathrm{cm\,yr^{-1}}$ for 2012–2017. The basal melt rate for these earlier years can,
however, not be estimated due to the lack of measurements. In the literature, basal melt rates have largely been attributed to
the ground heat flux and they are assumed to change only with the trend of the mean outside temperature. Luetscher et al.
(2008) emphasised that the ground heat flux is influenced by heat advected by water and air close to the cave walls to explain
the amount of observed basal melting. In recent years, the strong melting of the HIC ice body opened up and enlarged gaps
between the ice body and the surrounding rock, thus exposing a larger area of the cave ice to these heat fluxes and enabling air
and water to reach the bottom of the cave ice along new pathways, speeding up the melting process.

## 5  Conclusions

Despite not being a purely natural system due to show cave operations, HIC is a unique study object that provides the op-
portunity to extensively study the microclimate of sag-type ice caves. The long and spatially distributed continuous series of
temperature measurements inside the cave show a clear warming trend. In the main chamber, the calculated trend even ex-
ceeds the outside warming trend, rendering the cave and its perennial ice deposits particularly vulnerable to climate change.
External winter conditions strongly impact the cave temperature as they not only control the extent of cooling of the cave
atmosphere and the surrounding rock in winter but subsequently also influence the cave temperature in the following summer.
Hence, consistently cold winters would be necessary to maintain thermal conditions that are favourable for preserving cave ice
deposits. Observations, however, show a dramatic decrease in cave ice within the observation period. There is strong evidence
that, apart from melting at the surface, basal melting is another important factor at HIC. First efforts of relating the cave ice
development (ablation and total mass balance) with cave and outside air temperatures by applying a degree-day model show
promising results. The strong impact of the preceding winter on summer conditions enabled us to make a simplified prediction
of summer ablation based on winter temperature. Given the availability of calibration data, we are optimistic that this approach
is transferable to other sag-type ice caves. Further work is needed to better assess accumulation processes to improve the model
and eventually extend it to study the longer-term cave ice evolution. Overall, we have provided an extensive analysis of the
thermal conditions in HIC, whose perennial ice body is threatened by climate change and, if warming continues, is prone to
disappear within the next decades.

*Data availability.* Data are available from the authors upon request.

*Author contributions.* CS and FO carried out the measurements and provided the data. FO and MW designed the concept for manuscript.
TR prepared the cave map. MW performed the data analysis and wrote the manuscript with input and feedback from all co-authors.





*Competing interests.* The authors declare that they have no conflict of interest.

*Acknowledgements.* We thank Mathias Rotach for his valuable comments, the members of the local caving club (*Landesverein für Höh-*
*lenkunde Tirol*) for their continued interest and support, and the Austrian national weather service (ZAMG) and the Austrian Hydrological
Service for providing additional data. This project was supported by FWF grant P318740 and the Gottfried and Vera Weiss Prize to CS and
by additional funding by the Faculty of Geo- and Atmospheric Sciences of the University of Innsbruck.



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
