# Peer review of "Multi-annual temperature evolution and implications for cave ice development in a sag-type ice cave in the Austrian Alps"

_The Cryosphere, 2022_

## Author Comment (AC1)

**Author response to RC1**

Referee comment
Author response

GENERAL COMMENTS

Dear Editor,

I've read the manuscript "Multi-annual temperature evolution and implications for cave ice development in a sag-type ice cave in the Austrian Alps" by Wind et al.

I found the manuscript an interesting submission describing fully and comprehensively the microclimate of a sag-type ice cave. The manuscript fits with the purpose of the journal TC.

The manuscript reports significant information generally poorly or not addressed in the existing literature and it is, therefore, a valuable work.

Although pointed out several times and accurately described, the only "weakness" of the work relates to the lack of data calculating the impact of visitors in the cave, which is indeed something hard to quantify. This is not something that affects the quality of the paper itself but makes the findings a bit less important than what could have been achieved in a non-touristic cave.

Besides such general comments and the specific comments below, I suggest the manuscript can be published after minor revision.

Thank you very much for this encouraging review. We agree that the quantification of the impact of the cave management on the cave climate is a highly interesting point. However, other than making the reader aware of these influences, with the data we have we unfortunately cannot provide more information on this topic.

Please find the responses to the specific comments below.

SPECIFIC COMMENTS

**P 2 L 30-35**: as I agree with the statement "it is crucial to assess and understand the microclimatic and glaciological conditions inside ice caves and their coupling to the outside atmosphere" I suggest the innovative CFD model approach proposed by Bertozzi et al., (2019) "*On the interactions between airflow and ice melting in ice caves: A novel methodology based on computational fluid dynamics modelling" https://doi.org/10.1016/j.scitotenv.2019.03.074, 2019*is mentioned in this section.

We suggest to extend the text in the manuscript as follows (L39-41):

*Furthermore, the spatial distribution and temporal consistency of these measurements are mostly insufficient to allow comprehensive analyses of the full spatio-temporal characteristics. This also limits the validation of respective numerical models (e.g. Bertozzi et al., 2019).*

**Figure 1**: for more clarity, I suggest adding the location of the stakes even in the elevation view (lower panel)

We will adapt the figure accordingly.

**P 5 L 106** (also related to **P20 L 416-419**): I understood that, as you mentioned, it is really hard to quantify the effects of artificial snow input inside the cave, but can you be more specific about this process? I see that some information is retrievable from Fig. 8 and some are explained in the discussions but maybe you can add some more if known. For example: is the snow input affecting

all the areas homogeneously or just near the entrances, how often does it happen usually, just in late winter? Has the artificial snow input ever been quantified at least in snow thickness at a stake to have a vague idea of its impact (maybe referring to some of the Figure 8 values)? Is the shovelling process documented every time or the listed markers are just some of them?

The snow is brought in through the upper entrance and accumulates as a snow cone in the main ice-bearing chamber (Eisdom) as well as through the lower entrance where it fills the space below the staircase and feeds a secondary ice body. Figure 1 shows the areas with snow and the position of the stakes of which only Stake A (not further used in this study) is directly affected by artificial snow input.

Regarding the amount and the timing we can only work with respective notes by the local cavers. Thus, the markers in Figure 8 have been read from the guest book of the hut next to the cave documenting the timing of artificial snow input into the cave. This information is reliable, but the quantity of snow input was never documented.

We suggest to add the following sentences in the text (L 106): "*The snow is brought in through the upper entrance accumulating as a snow cone in the main ice-bearing chamber (Eisdom) as well as through the lower entrance where it fills the space below the staircase and feeds a secondary ice body (Fig. 1). Although these activities are documented, proper quantification of the effect of the artificial snow input on the cave ice mass balance is not feasible. Regarding stake measurements, only stake A was directly affected by the artificial snow input and thus not used in this study.*"

**P21 L 430-437**: I feel that having a range of values from other stakes and T sensors would enrich the discussions of this work and improve the eventual future comparisons with other studies using this methodology in different caves. I understand that stake B and T29 were used as references for deriving the DDF as they are more robust. Is there a chance that some other T sensors and stakes are used for calculation of shorter DDF periods and then compared with the reference values that you already mentioned? If stake B is affected by the artificial snow input, are there other stakes that can be less affected by snow shovelling and therefore can provide additional data in the discussion of DDF findings?

The combination of stake B with logger T29 was chosen not only because it is the longest continuous series, but also because the two measuring points are closest to each other (~2 m distance). Moreover, stake B was rarely affected by the artificial snow input as is known from the regular readings in spring an autumn. Following the comment we report the so far not shown degree day factors using other stake-logger combinations (Table 1). The values range from 0.6 mm $°C^{-1} day^{-1}$ up to 5.9 mm $°C^{-1} day^{-1}$ with the highest values resulting from combinations with T36. This temperature logger is furthest away from the stake measurements and represents a thermal regime which is less relevant for ice developments in the main chamber.

Table 1: Degree day factor (DDF) in mm $°C^{-1} day^{-1}$ calculated from different logger-stake combinations (see Figure 8).

| logger \ stake | A | B | D | E |
|---|---|---|---|---|
| T29 | 2.2 | 1.9 | 1.4 | 0.6 |
| T30 | 1.9 | 2.2 | 1.7 | 0.9 |
| T36 | 5.9 | 5.1 | 4.0 | 1.7 |

---

## Author Comment (AC2)

**Author response to RC2 (Aurel Perşoiu)**

Referee comment
    Author response

First of all we would like to thank the reviewer for his thorough review and valuable comments. Please see our responses below.

7-8: this sentence is quite uninformative

We suggest to change the text as follows: "*The open period is characterised by unstable to neutral stratification which is an effect of convection during episodes when cold air can penetrate into the cave. Criteria to detect corresponding periods are investigated.*"

26: "trap" would suffice

We will change "thermal trap" to "trap" in the text.

39-40: it is not clear how this sentence is linked to the case study. "Comprehensive" analysis for several caves or for this one only? It could be safely left out.

Linked to the previous sentence (line 37-39) we wanted to point out that, apart from the general lack of long-term measurements, the spatial coverage of temperature measurements in caves is usually limited to only a few loggers per cave and thus does not enable a detailed analysis of the spatial temperature patterns inside a cave.

We suggest to change the respective sentence to: "*Furthermore, the spatial distribution and temporal consistency of these measurements are mostly insufficient to allow comprehensive analyses of the full spatio-temporal characteristics.*"

41: perhaps "ice level dynamics" (or similar) instead of "stake" records?

"Stake record" is often used in the glaciological literature and we find it appropriate in the context of this work, too.

We suggest to change the text to: "*We aim to fill this gap by analysing long-term data (2008 to 2021) from a network of temperature logger and ablation stakes at a sag-type ice cave in the Austrian Alps, Hundsalm ice cave.*"

61: the diameters of the entrance shafts could be an important information for air circulation, please add them if available

The entrances measure about 3 x 8 m (lower entrance) and 3 x 4 m (upper entrance). This information will be added to the manuscript.

76: please give the distance between the air measurement point and the nearest ice body. It is helpful to interpret air temperature variability and role of latent heat in shaping it.

T29 was initially mounted ~1 m above the ice surface. This distance increased over the years with the decrease of the ice surface to ~2 m in 2021.

T30 is ~10 m away from the ice surface, and T36 is 1.5 m above the ice.

96: what is the altitude of the precipitation sampling site?

Buchacker station is at an elevation of 1425 m above sea level. We will add this information to the manuscript.

104: does this shoveled snow reaches areas where air temperature ad/or ice dynamics are monitored?

The snow is mainly shoveled into the main chamber (Eisdom) and below the staircase along the lower entrance. Stake A would have been influenced the most by these activities as it was in the center of the snow cone below the upper entrance but it got damaged/removed before snow was shoveled in on a regular (yearly) basis. Only a few times the shoveled snow reached stake B. In these cases the measurements were still taken at the ice surface, removing the extra snow on top. The same is true for P4. Stakes at other parts of the cave were not directly affected (see also reply to the comment to P5 L 106 of RC1).

We suggest to extend the text (L 106) as follows: "*The snow is brought in through the upper entrance forming a snow cone in the main ice-bearing chamber (Eisdom) as well as through the lower entrance where it fills the space below the staircase and feeds a secondary ice body (see Fig. 1). Although these activities are documented, proper quantification of the effect of the artificial snow input on the cave ice mass balance is not feasible. Regarding stake measurements, only stake A was directly affected by the artificial snow input and thus not used in this study.*"

Regarding the effects of the snow shoveling on temperature, we did not see any direct influence of these activities in the temperature record. As already stated in the manuscript, a more detailed assessment of the overall effect of the artificial snow input on the cave temperature can unfortunately not be made due to the lack of respective data.

105: somewhat strange, perhaps the climate is manipulated, not the entire cave?

We argue that the cave itself has been manipulated (e.g. by adding a door at the lower entrance, and an air lock at the passage to the lower (ice-free) part of the cave) and these changes as well as other activities (shoveling of snow, tourists) have some impact on the cave climate. However, we think that the main characteristics of the cave climate are largely unaffected by human interference.

122: how does this filtering influences the long-term averages calculated below?

The difference in the mean annual temperature (Table 1) compared to the unfiltered time series is < 0.01 °C. For the averages from May to October, the period most affected by the filtering, the difference is < 0.02 °C (see L 121-122).

131: this could be very useful for any subsequent studies. However, while deriving potential temperature from pressure data is quite straightforward in the free atmosphere, it might prove problematic in cave settings due to potential biases induced by pressure changes linked to movement of air inside cave passages. Did you consider these, and also potential differences between summer and winter?

Since we only have pressure measurements at the outside station we do not have the data to check whether dynamic effects on air pressure have significant influence on the results.

According to meanwhile performed measurements, air velocity at a location close to T29 hardly exceeds 0.5 m s$^{-1}$. Respectively induced dynamic pressure changes are negligible in this context (< 1 Pa compared to $10^3$ Pa).

147: normalized?

we will change "normed" to "normalised" in the text.

161-162: the warming trend is quite interesting, and puzzling, all the same. While it is tempting to see it as a sign of a warming climate, the fact that the logger located in the non-glaciated part of

the cave does not register it (nor the external one) makes one wonder if the trend is related perhaps to changing distance from ice. melting of ice would necessarily act as a heat sink, thus keeping the temperature of air in the nearby atmosphere at 0 °C as long as ice is present. Any additional hat added to the air (by, e.g., warming outside) would be used to melt additional ice and thus removing any increase. So, how far from the ice are the loggers showing the warming trend placed? Did this distance increase? Did you detect any breakpoint in the time series linked to, e.g., drop in ice level?

We show that there is also a warming trend in the non-glaciated part of the cave (0.024 °C yr$^{-1}$, line 166-167) which is similar to the trend at the lowest point in the ice-bearing part of the cave (T36: +0.027 °C yr$^{-1}$).

We explain the "missing" trend outside by the relatively short period for trend calculation with the much higher outside temperature variability compared to the cave temperature. For a longer time series at a highly correlated station (Hahnenkamm, located approx. 30 km south-east of Hundsalm at an elevation of 1794 m) a statistically significant trend was found. We address this issue in the discussion (lines 346-348).

The distance between the ice surface and the air temperature measurement at T29, where we see the strongest trend, increased from ~1 m to 2 m (which is in agreement with the measurements at stake B). As this increase is gradual, we did not detect any breakpoint in the time series.

The distance to the ice body did not change significantly at T36 and the area around T30 was always ice free.

179: how was this threshold chosen?

The threshold was chosen empirically by analysing the time series of daily standard deviation values and finding a value that was exceeded regularly during cold air intrusions at all three long-term cave monitoring sites. A similar criterion was used by Racine et al. (2022).

Racine, T., Spötl, C., Reimer, P. & Čarga, J. (2022). Radiocarbon constraints on periods of positive cave ice mass balance during the last millennium, Julian Alps (NW Slovenia). Radiocarbon, 64(2), 333-356. doi:10.1017/RDC.2022.26

189-192 (and lines above): I find the discussion on the net external cooling required to induce a net cave cooling interesting and stimulating. Especially intriguing are the values of the net differences between outside and inside which are quite high (8.5 °C!). Perhaps daily means are masking the real difference, as minima tend to occur at different times in and out of the cave? Did you try a cross-correlation analysis that would indicate the time lag between external and internal variations and thus help sustain these very large differences? Perrier et al. (2005) for instances found very short times for cold air "avalanches„ reaching lower parts of caves

We tried cross-correlation for episodes with cold air intrusions but found that in most cases the 2h measuring interval was too coarse to see a phase shift (see lines 387-389), meaning that these "cold air avalanches" reach lower parts of the cave within the 2h interval (see Fig. 6 and related discussions). However, the amplitude of the temperature signal gets damped with increasing depth and hence for the loggers deeper in the cave to exceed the defined threshold of $\sigma > 0.1$ °C the cold air intrusion has to be correspondingly stronger (i.e. the difference between outside and inside air has to be higher).

202-203: this is an important observation, yet difficult to reconciliate with physics. Basically, the ms says that weak cooling in winter somehow results in warmer summers. Now, in any system where a heat sink is present (melting ice, in this case), temperature will be controlled by latent heat.

Further, the rock surrounding the cave has an oversized fingerprint on the overall thermal balance of the cave air+cave ice system. In the absence of the meting ice, one could imagine that weaker cooling in winter leads to warmer summer air temperatures, but the melting of ice would obliterate any such influence. Basically, you should provide a mechanistic explanation for the processes that lead from weak winter cooling to warmer summers – this would be a major point for future similar studies.

Our data clearly shows that there is a strong correlation between winter and subsequent summer air temperatures. Latent heat due to the melting of ice dampens the summer warming to some extent but in the case of HIC cannot stop the cave from warming above 0 °C (not even at the lowest logger T36 surrounded by ice). If there would be much more ice present in the cave, the impact of latent heat would possibly be higher. However, we would still expect a link between winter and subsequent summer temperature as the amount of winter cooling also influences ice accumulation and subsequently the amount of ice available for melting during summer.

We are currently working on a more physically based explanation for this relationship that is planned to be submitted elsewhere.

Chapter 3.3. This is a long chapter with very detailed discussion of the data that seems to result in a loss of focus. Perhaps the data description should be shortened and the discussion focus on the interaction between cold air intrusion, distance of air measurements points from ice and the role of internal air circulation. These are all linked and the presence of ice acts as a strong modifier of air circulation/temperature. This could/should perhaps merged with the subsequent chapter 3.4 (which I will not discuss further down).

We think that this chapter is an important part of this manuscript as it gives detailed insights into the vertical air temperature profile, its temporal and spatial variability and implications for cave ventilation.

Chapter 3.5. The discussion of rock/ice temperatures could be used to support/reject the inferences made on lines 202-203 (see above).

We will consider adapting the discussion.

Chapter 3.6. I miss a discussion of the links between PDD outside the cave and ice dynamics – this would help understand the role of external air temperature variations on ice dynamics – see also the opening line of the discussions (L343)

We found that there is no significant correlation between PDD outside the cave and ice dynamics. This can be explained by the fact that the outside air is largely decoupled from the cave atmosphere during the ablation period due to the strong stable stratification that prohibits air exchange. This is why we used the FDD outside the cave for the model, as the cave atmosphere mainly interacts with the outside atmosphere during cold air intrusions in the accumulation period.

We will adapt the corresponding text in the manuscript to make this clearer.

12 – well, this lack of correlation is somehow normal. Dripping water, direct snowfall and snow shoveling by cave managers result in a complex and possibly impossible to understand link between snow accumulation and precipitation amount.

We agree with this comment.

315-319: I am not sure a model that excludes outside temperature would help understand the ice dynamics, this should be included.

The third model in this list uses outside temperature (freezing degree days during the accumulation period) as the predictor. Since there is no correlation of ablation or mass balance with outside PDD we did not include such a model.

335 – this density refers to ice at maximum density. Is this the case here? I would expect lower density, based on how ice forms.

This is a good point. We repeated our calculations using the density value of 870 kg m$^{-3}$ found in a study in Eisriesenwelt (May et al, 2011) and will adapt the text accordingly.

May, B., Spötl, C., Wagenbach, D., Dublyansky, Y., and Liebl, J.: First investigations of an ice core from Eisriesenwelt cave (Austria), The Cryosphere, 5, 81–93, https://doi.org/10.5194/tc-5-81-2011, 2011.

338 – these are extremely high values. What are the errors associated to the measurements?

The error of the ice measurements is expected to be ± 1 cm .

344-346 – this is extremely interesting, but perhaps it should be moved after the discussion of the data.

351-358 – this section somehow does not fit well in here, especially given the strong opening statement of the section (344-346)

We agree with these two comments and will rearrange this part of the discussion.

372 and subsequent: again, apart from correlation, which can be the result of artifacts in statistical analyses, an explanation is required. Basically here, the results are presented again but no discussion follows.

We will move parts of the discussion, including Fig. 10, to the results section and expand the discussion on this subject. However, we do not think that the correlation between winter and subsequent summer temperature is a statistical artifact. As a test we tried to do the same correlation the other way around (with summer and subsequent winter temperatures) and found no correlation at all.

412-415: again, see my comments above. Melting in summer has to be the result of warm temperatures and/or the sum of low winter accumulation and (high) summer melting, rather than warm winters only. Also, the unquantified snow shoveling must play an (oversized) role.

The summer melt rate is related to the respective available energy during melting and thus is related to the summer conditions which are influenced by the preceding winter conditions. We think our data provides sufficient evidence to support this conclusion.

General observation for the "discussions" section: this study can be broken down on a climate analysis and links between ice dynamics and climate. The first part is nicely done, however, the links with ice dynamics are somehow weakly supported by the observations and hampered by the anthropic influence. I suggest reducing the entire discussion to the discussion of 1) cave climate and 2) links with ice, but with the later stating from the beginning the fact that snow shoveling inside the cave strongly masks the natural processes.

Apart from the suggested changes mentioned in the previous comments (i.e. moving Fig. 10 to the results section and rearranging chapter 4.1), we want to keep the structure of the rest of the discussion as it is now. Furthermore, we already pointed out the potential influence of snow shoveling in the discussion (line 416-422).

---

## Author Response (AR2)

Referee 1 comment (line numbers refer to the originally submitted manuscript)
Referee 2 comment
Author response
*Adapted text in the manuscript (pages and line numbers refer to the revised version of the manuscript)*

L7-8: this sentence is quite uninformative

*P1 L6-7: The open period is characterised by unstable to neutral stratification which is an effect of convection during episodes when cold air can penetrate into the cave. Criteria to detect corresponding periods are investigated.*

L26: "trap" would suffice

We changed "*thermal trap*" to "*trap*" in the text.

*P2 L28: In the summer months (closed period) the cave atmosphere is largely decoupled from the outside while in the winter months (open period) the cave serves as a trap for cold air (Perşoiu, 2018).*

L30-35: As I agree with the statement "it is crucial to assess and understand the microclimatic and glaciological conditions inside ice caves and their coupling to the outside atmosphere" I suggest the innovative CFD model approach proposed by Bertozzi et al., (2019) "*On the interactions between airflow and ice melting in ice caves: A novel methodology based on computational fluid dynamics modelling*" https://doi.org/10.1016/j.scitotenv.2019.03.074, 2019 is mentioned in this section.

L39-40: it is not clear how this sentence is linked to the case study. "Comprehensive" analysis for several caves or for this one only? It could be safely left out.

Linked to the previous sentence we wanted to point out that, apart from the general lack of long-term measurements, the spatial coverage of temperature measurements in caves is usually limited to only a few loggers per cave and thus does not enable a detailed analysis of the spatial temperature patterns inside a cave.

*P2 L40-42: Furthermore, the spatial distribution and temporal consistency of these measurements are mostly insufficient to allow comprehensive analyses of the full spatio-temporal characteristics. This also limits the validation of respective numerical models (e.g. Bertozzi et al., 2019).*

L41: perhaps "ice level dynamics" (or similar) instead of "stake" records?

"Stake record" is often used in the glaciological literature and we find it appropriate in the context of this work, too.

*P2 L43-44: We aim to fill this gap by analysing long-term data (2008 to 2021) from a network of temperature logger and ablation stakes at a sag-type ice cave in the Austrian Alps, Hundsalm ice cave.*

Figure 1: for more clarity, I suggest adding the location of the stakes even in the elevation view (lower panel)

We adapted Figure 1 accordingly (P4).

L61: the diameters of the entrance shafts could be an important information for air circulation, please add them if available

The entrances measure about 3 x 8 m (lower entrance) and 3 x 4 m (upper entrance).

*P3 L64: The upper entrance is a 25 m high shaft (3 m x 4 m wide), while the lower entrance (3 m x 8 m wide), which opens a few meters below the upper one, serves as the main entrance and is equipped with a staircase as well as a door (15 m below the upper entrance) that is closed during summer and fall.*

L76: please give the distance between the air measurement point and the nearest ice body. It is helpful to interpret air temperature variability and role of latent heat in shaping it.

T29 was initially mounted ~1 m above the ice surface. This distance increased over the years with the decrease of the ice surface to ~2 m in 2021. T30 is ~10 m away from the ice surface and T36 is 1.5 m above the ice.

*P3 L79-81: T29 was initially mounted ~1 m above the ice surface. This distance increased over the years with the decrease of the ice surface to ~2 m in 2021. T30 is ~10 m away from the ice surface and T36 is 1.5 m above the ice.*

L96: what is the altitude of the precipitation sampling site?

Buchacker station is at an elevation of 1425 m above sea level.

*P5 L100-102: Therefore, monthly precipitation sums from a totalisator operated by the Austrian Hydrological Service (station Buchacker, located less than 2 km south-west of the cave at 1425 m above sea level) were used.*

L104: does this shoveled snow reaches areas where air temperature ad/or ice dynamics are monitored?

L106 (also related to P20 L416-419): I understood that, as you mentioned, it is really hard to quantify the effects of artificial snow input inside the cave, but can you be more specific about this process? I see that some information is retrievable from Fig. 8 and some are explained in the discussions but maybe you can add some more if known. For example: is the snow input affecting all the areas homogeneously or just near the entrances, how often does it happen usually, just in late winter? Has the artificial snow input ever been quantified at least in snow thickness at a stake to have a vague idea of its impact (maybe referring to some of the Figure 8 values)? Is the shovelling process documented every time or the listed markers are just some of them?

The snow is mainly shoveled into the main chamber (Eisdom) and below the staircase along the lower entrance. Stake A would have been influenced the most by these activities as it was in the center of the snow cone below the upper entrance but it got damaged/removed before snow was shoveled in on a regular (yearly) basis. Only a few times the shoveled snow reached stake B. In these cases the measurements were still taken at the ice surface,

removing the extra snow on top. The same is true for P4. Stakes at other parts of the cave were not directly affected.

Regarding the amount and the timing we can only work with respective notes by the local cavers. Thus, the markers in Figure 8 have been read from the guest book of the hut next to the cave documenting the timing of artificial snow input into the cave. This information is reliable, but the quantity of snow input was never documented.

*P5 L112-117: The snow is brought in through the upper entrance accumulating as a snow cone in the main ice-bearing chamber (Eisdom) as well as through the lower entrance where it fills the space below the staircase and feeds a secondary ice body (Fig. 1). Although these activities are documented, proper quantification of the effect of the artificial snow input on the cave ice mass balance is not feasible. Regarding stake measurements, only stake A was directly affected by the artificial snow input and thus not used in this study.*

**L105: somewhat strange, perhaps the climate is manipulated, not the entire cave?**

We argue that the cave itself has been manipulated (e.g. by adding a door at the lower entrance, and an air lock at the passage to the lower (ice-free) part of the cave) and these changes as well as other activities (shoveling of snow, tourists) have some impact on the cave climate. However, we think that the main characteristics of the cave climate are largely unaffected by human interference.

**L122: how does this filtering influences the long-term averages calculated below?**

The difference in the mean annual temperature (Table 1) compared to the unfiltered time series is < 0.01 °C. For the averages from May to October, the period most affected by the filtering, the difference is < 0.02 °C (see P6 L131-132).

**L131: this could be very useful for any subsequent studies. However, while deriving potential temperature from pressure data is quite straightforward in the free atmosphere, it might prove problematic in cave settings due to potential biases induced by pressure changes linked to movement of air inside cave passages. Did you consider these, and also potential differences between summer and winter?**

Since we only have pressure measurements at the outside station we do not have the data to check whether dynamic effects on air pressure have significant influence on the results.

According to meanwhile performed measurements, air velocity at a location close to T29 hardly exceeds 0.5 m s$^{-1}$. Respectively induced dynamic pressure changes are negligible in this context (< 1 Pa compared to $10^3$ Pa).

**L147: normalized?**

*P6 L147-158: Finally, degree-day sums were normalised by the number of days in each period.*

**L161-162: the warming trend is quite interesting, and puzzling, all the same. While it is tempting to see it as a sign of a warming climate, the fact that the logger located in the non-glaciated part of**

the cave does not register it (nor the external one) makes one wonder if the trend is related perhaps to changing distance from ice. melting of ice would necessarily act as a heat sink, thus keeping the temperature of air in the nearby atmosphere at 0 °C as long as ice is present. Any additional hat added to the air (by, e.g., warming outside) would be used to melt additional ice and thus removing any increase. So, how far from the ice are the loggers showing the warming trend placed? Did this distance increase? Did you detect any breakpoint in the time series linked to, e.g., drop in ice level?

We show that there is also a warming trend in the non-glaciated part of the cave (0.024 °C yr$^{-1}$, line 166-167) which is similar to the trend at the lowest point in the ice-bearing part of the cave (T36: +0.027 °C yr$^{-1}$).

We explain the "missing" trend outside by the relatively short period for trend calculation with the much higher outside temperature variability compared to the cave temperature. For a longer time series at a highly correlated station (Hahnenkamm, located approx. 30 km south-east of Hundsalm at an elevation of 1794 m) a statistically significant trend was found. We address this issue in the discussion (L368-372).

The distance between the ice surface and the air temperature measurement at T29, where we see the strongest trend, increased from ~1 m to 2 m (which is in agreement with the measurements at stake B). As this increase is gradual, we did not detect any breakpoint in the time series.

The distance to the ice body did not change significantly at T36 and the area around T30 was always ice free.

*P19 L371-380: The presented results paint a comprehensive picture of a sag-type ice cave that is threatened by increasing temperatures, indicated by a significant warming trend in all parts of the cave. The trend in the non-glaciated part of the cave (+0.024 °C yr$^{-1}$) compares well to the trend in the lowest part of the ice-bearing part of the cave (T36: +0.027 °C yr$^{-1}$). At the cave site with the strongest trend (T29) the distance between the ice surface and the air temperature measurement increased by approximately 1 m over the observation period (verified by measurements at stake B) while at the other main cave monitoring locations the distance to the ice stayed roughly the same. This increasing distance of T29 to the ice surface possibly enhances the trend at this location. However, as this increase is gradual, we did not detect any break point in the T29 time series due to e.g. a rapid drop in ice level. For the outside temperature the Mann–Kendall test did not yield a significant trend. This is most probably caused by the relatively short time series used for trend calculations. Using a longer time series (1993–2021) of the highly correlated ZAMG station Hahnenkamm results in a trend of +0.043 °C yr$^{-1}$. Assuming that the long-term trend at our cave site is similar means that the warming trend in the main chamber of the cave (+0.054 °C yr$^{-1}$) is even larger compared to the outside trend.*

L179: how was this threshold chosen?

*P8 L196-198: The threshold was chosen empirically by analysing the time series of daily standard deviation values and finding a value that was exceeded regularly during cold air intrusions at all three long-term cave monitoring sites. A similar criterion was used by Racine et al. (2022).*

L189-192 (and lines above): I find the discussion on the net external cooling required to induce a net cave cooling interesting and stimulating. Especially intriguing are the values of the net differences between outside and inside which are quite high (8.5 °C!). Perhaps daily means are masking the real difference, as minima tend to occur at different times in and out of the cave? Did you try a cross-correlation analysis that would indicate the time lag between external and internal variations and thus help sustain these very large differences? Perrier et al. (2005) for instances found very short times for cold air "avalanches„ reaching lower parts of caves

We tried cross-correlation for episodes with cold air intrusions but found that in most cases the 2h measuring interval was too coarse to see a phase shift (see lines 387-389), meaning that these "cold air avalanches" reach lower parts of the cave within the 2h interval (see Fig. 6 and related discussions). However, the amplitude of the temperature signal gets damped with increasing depth and hence for the loggers deeper in the cave to exceed the defined threshold of $\sigma > 0.1$ °C the cold air intrusion has to be correspondingly stronger (i.e. the difference between outside and inside air has to be higher).

L202-203: this is an important observation, yet difficult to reconciliate with physics. Basically, the ms says that weak cooling in winter somehow results in warmer summers. Now, in any system where a heat sink is present (melting ice, in this case), temperature will be controlled by latent heat. Further, the rock surrounding the cave has an oversized fingerprint on the overall thermal balance of the cave air+cave ice system. In the absence of the meting ice, one could imagine that weaker cooling in winter leads to warmer summer air temperatures, but the melting of ice would obliterate any such influence. Basically, you should provide a mechanistic explanation for the processes that lead from weak winter cooling to warmer summers – this would be a major point for future similar studies.

Our data clearly shows that there is a strong correlation between winter and subsequent summer air temperatures. Latent heat due to the melting of ice dampens the summer warming to some extent but in the case of HIC cannot stop the cave from warming above 0 °C (not even at the lowest logger T36 surrounded by ice). If there would be much more ice present in the cave, the impact of latent heat would possibly be higher. However, we would still expect a link between winter and subsequent summer temperature as the amount of winter cooling also influences ice accumulation and subsequently the amount of ice available for melting during summer.

We are currently working on a more physically based explanation for this relationship that is planned to be submitted elsewhere.

*P20 L388-391: Although latent heat used for the melting of ice dampens the summer warming inside the cave to some extent, it is not sufficient to stop the cave from warming above 0 °C. However, the amount of winter cooling has an influences on ice accumulation which subsequently determines the amount of ice available for melting (and cooling the cave) during summer.*

Chapter 3.5. The discussion of rock/ice temperatures could be used to support/reject the inferences made on lines 202-203 (see above).

In mild winters with only short cold air intrusions the surrounding rock is hardly cooled due to the phase shift and the dampening of the air temperature signal. Only long cold phases can significantly cool down a thicker layer of the rock. Hence, after a consistently cold winter the rock can serve as a heat sink during the transition period in early spring while a heat flux

directed from the deeper (warmer) layers of the rock towards the cave atmosphere is providing heat to the cave system during the rest of the year.

Rock temperature furthermore plays a role in the availability of seepage water for refreezing. Due to the phase shift with air temperature, phases with positive rock temperature and negative cave air temperature provide ideal conditions for ice accumulation through refreezing of seepage water.

Chapter 3.6. I miss a discussion of the links between PDD outside the cave and ice dynamics – this would help understand the role of external air temperature variations on ice dynamics – see also the opening line of the discussions (L343)

We found that there is no significant correlation between PDD outside the cave and ice dynamics. This can be explained by the fact that the outside air is largely decoupled from the cave atmosphere during the ablation period due to the strong stable stratification that prohibits air exchange. This is why we used the FDD outside the cave for the model, as the cave atmosphere mainly interacts with the outside atmosphere during cold air intrusions in the accumulation period.

*P16 L314-317: No significant correlation was found with PDD of $T_{out}$. Nevertheless, the correlation of ablation with cave stations provides the opportunity to study the potential of a degree-day model, commonly used in glacier mass balance studies (e.g., Hock, 2003; Kuhn et al., 1999), that quantitatively relates ablation to the temperature sum, inside a cave.*

L312 – well, this lack of correlation is somehow normal. Dripping water, direct snowfall and snow shoveling by cave managers result in a complex and possibly impossible to understand link between snow accumulation and precipitation amount.

We agree with this comment.

L315-319: I am not sure a model that excludes outside temperature would help understand the ice dynamics, this should be included.

The third model in this list uses outside temperature (freezing degree days during the accumulation period) as the predictor. Since there is no correlation of ablation or mass balance with outside PDD we did not include such a model.

*P17 L329: Since the PDD of $T_{out}$ yielded no correlation with ablation, the freezing degree-day sum (FDD) of the preceding accumulation period at the outside station was used (correlation value r = -0.8) and yields a DDF of 0.5 mm °C$^{-1}$ day$^{-1}$ .*

L335: this density refers to ice at maximum density. Is this the case here? I would expect lower density, based on how ice forms.

This is a good point. We repeated our calculations using the density value of 870 kg m$^{-3}$ found in a study in Eisriesenwelt (May et al, 2011) and will adapt the text accordingly.

*P19 L358-359: We assumed an effective thermal conductivity of homogeneous limestone of $\lambda = 2$ W m$^{-1}$ K$^{-1}$ and a density of ice of $\rho_{ice} = 870$ kg m$^{-3}$ (May et al., 2011).*

*P19 L363-364: The degree-day method as well as Eq. 3 using T36 and T50r yield comparable results of 9.1 cm and 8.7 cm, respectively. Finally, Eq. (3) applied to the rock temperatures gives the lowest value of 3.3 cm.*

L338 : these are extremely high values. What are the errors associated to the measurements?

The error of the ice measurements is expected to be ± 1 cm .

*P5 L103-104: The cave ice development has been monitored using stakes placed in different parts of the ice body that were measured manually at least twice a year starting in summer 2007 (Fig. 1, upper panel) with an estimated accuracy of ± 1 cm.*

L344-346 – this is extremely interesting, but perhaps it should be moved after the discussion of the data.

L351-358 – this section somehow does not fit well in here, especially given the strong opening statement of the section (344-346)

We agree with these two comments and partly rearranged the discussion (Section 4.1, P19-20 L366-395).

L372 and subsequent: again, apart from correlation, which can be the result of artifacts in statistical analyses, an explanation is required. Basically here, the results are presented again but no discussion follows.

We moved the left panel of Fig. 10 (now Fig. 3) and corresponding text (monthly mean correlation outside vs. inside) to chapter 3.1.

*P8 L186-192: A seasonal pattern was also observed in the correlation between outside and cave temperature. From May to September the correlation between T out and Eisdom (T29) (monthly Pearson correlation r of 2 h data, 01 June 2008 to 31 March 2021) drops below a value of 0.2 as the outside air is decoupled from the cave atmosphere due to the prevailing stable stratification (Fig. 3). In contrast, the period from November to March shows the strongest correlation with values above 0.6 and a maximum correlation of 0.77 in February. Tiefster Punkt (T36) and Tropfsteinhalle (T30) show a similar seasonal pattern but weaker correlations during the open period (maximum correlation in February of 0.65 at T36 and 0.57 at T30) as a result of the greater distance to the cave entrance compared to T29.*

We moved the right panel of Fig. 10 (now Fig. 5) and corresponding text (linear regression of temperature sums) to chapter 3.2:

*P11 L228-L232: Temperature sums are primarily used to model the ablation or the mass balance of ice (Section 3.6), but we use them to study the potential relationship of winter temperature inside the cave on the preceding summer characteristics. Figure 5 shows that a clear linear relationship can be established between the accumulated temperature during the winter half-year (November to April) and the subsequent summer temperature sum (May to October). The slope of the regression line varies between the logger locations with T30 showing the steepest and T36 the lowest incline.*

However, we do not think that the correlation between winter and subsequent summer temperature is a statistical artifact. As a test we tried to do the same correlation the other way around (with summer and subsequent winter temperatures) and found no correlation at all.

L412-415: again, see my comments above. Melting in summer has to be the result of warm temperatures and/or the sum of low winter accumulation and (high) summer melting, rather than warm winters only. Also, the unquantified snow shoveling must play an (oversized) role.

The summer melt rate is related to the respective available energy during melting and thus is related to the summer conditions which are influenced by the preceding winter conditions. We think our data provides sufficient evidence to support this conclusion.

L430-437: I feel that having a range of values from other stakes and T sensors would enrich the discussions of this work and improve the eventual future comparisons with other studies using this methodology in different caves. I understand that stake B and T29 were used as references for deriving the DDF as they are more robust. Is there a chance that some other T sensors and stakes are used for calculation of shorter DDF periods and then compared with the reference values that you already mentioned? If stake B is affected by the artificial snow input, are there other stakes that can be less affected by snow shovelling and therefore can provide additional data in the discussion of DDF findings?

The combination of stake B with logger T29 was chosen not only because it is the longest continuous series, but also because the two measuring points are closest to each other (~2 m distance). Moreover, stake B was rarely affected by the artificial snow input as is known from the regular readings in spring an autumn. Following the comment we report the so far not shown degree day factors using other stake-logger combinations (Table 1). The values range from 0.6 mm $°C^{-1}$ $day^{-1}$ up to 5.9 mm $°C^{-1}$ $day^{-1}$ with the highest values resulting from combinations with T36. This temperature logger is furthest away from the stake measurements and represents a thermal regime which is less relevant for ice developments in the main chamber.

Table 1: Degree day factor (DDF) in mm $°C^{-1}$ $day^{-1}$ calculated from different logger-stake combinations for the available periods (see Figure 8).

| logger \ stake | A | B | D | E |
|---|---|---|---|---|
| T29 | 2.2 | 1.9 | 1.4 | 0.6 |
| T30 | 1.9 | 2.2 | 1.7 | 0.9 |
| T36 | 5.9 | 5.1 | 4.0 | 1.7 |

General observation for the "discussions" section: this study can be broken down on a climate analysis and links between ice dynamics and climate. The first part is nicely done, however, the links with ice dynamics are somehow weakly supported by the observations and hampered by the anthropogenic influence. I suggest reducing the entire discussion to the discussion of 1) cave climate and 2) links with ice, but with the later stating from the beginning the fact that snow shoveling inside the cave strongly masks the natural processes.

Apart from the suggested changes mentioned in the previous comments (i.e. moving Fig. 10 to the results section and partially changing chapter 4.1), we want to keep the structure of the rest of the discussion as it is now. Furthermore, we already pointed out the potential influence of snow shoveling in the discussion (P21 L433-438).